# Nitrogen loss processes in response to upwelling in a Peruvian coastal setting dominated by denitrification – a mesocosm approach

Kai G. Schulz[1], Eric P. Achterberg[2], Javier Arístegui[3], Lennart T. Bach[4], Isabel Baños[3], Tim Boxhammer[2], Dirk Erler[1], Maricarmen Igarza[5], Verena Kalter[2,6], Andrea Ludwig[2], Carolin Löscher[7], Jana Meyer[2], Judith Meyer[2], Fabrizio Minutolo[2], Elisabeth von der Esch[8], Bess B. Ward[9], and Ulf Riebesell[2]

[1]Centre for Coastal Biogeochemistry, School of Environment, Science and Engineering, Southern Cross University, Lismore, NSW, Australia
[2]GEOMAR Helmholtz Centre for Ocean Research Kiel, Kiel, Germany
[3]Instituto de Oceanografía y Cambio Global, IOCAG, Universidad de Las Palmas de Gran Canaria ULPGC, Las Palmas, Spain
[4]Institute for Marine and Antarctic Studies, University of Tasmania, Hobart, Tasmania, Australia
[5]Instituto del Mar del Perú (IMARPE), Dirección General de Investigaciones en Oceanografía y Cambio Climático, Callao, Peru
[6]Memorial University of Newfoundland, Department of Ocean Sciences, Logy Bay, Newfoundland, Canada
[7]Nordcee, DIAS, Department of Biology, University of Southern Denmark, Campusvej 55, Odense M-DK
[8]Institute of Hydrochemistry, Chair of Analytical Chemistry and Water Chemistry, Technical University of Munich, Munich, Germany
[9]Department of Geosciences, Princeton University, Princeton, New Jersey 08544

**Correspondence:** Kai G. Schulz (kai.schulz@scu.edu.au)

**Abstract.** Upwelling of nutrient-rich deep waters make Eastern Boundary upwelling systems (EBUS), such as the Humboldt Current System, hotspots of marine productivity. Associated settling of organic matter to depth and consecutive aerobic decomposition results in large sub-surface water volumes being oxygen-depleted. Under these circumstances, organic matter remineralisation can continue via denitrification which represents a major loss pathway of bioavailable nitrogen. Additionally, anaerobic ammonium oxidation can remove significant amounts of nitrogen in these areas. Here we assess the interplay of sub-oxic water upwelling and nitrogen cycling in a manipulative off-shore mesocosm experiment. Measured denitrification rates in incubations with water from the oxygen-depleted bottom layer of the mesocosms, via $^{15}$N label incubations, mostly ranged between 5.5–20 (interquartile range), reaching up to 80 $\mathrm{nmol\,N_2\,L^{-1}h^{-1}}$. However, actual in-situ rates in the mesocosms, estimated via Michaelis-Menten kinetic scaling, did most likely not exceed 0.2–4.2 $\mathrm{nmol\,N_2\,L^{-1}h^{-1}}$ (interquartile range), due to substrate limitation. In the surrounding Pacific measured denitrification rates were similar, although indications of substrate limitation were detected only once. Both in the mesocosms and the Pacific Ocean anammox made only a minor contribution to overall nitrogen loss when encountered, potentially related to organic matter C/N stoichiometry and/or process-specific oxygen and hydrogen sulphide sensitivities. Over the first 38 days of the experiment, total nitrogen loss calculated from in-situ rates of denitrification and anammox was comparable to estimates from a full nitrogen budget in the mesocosms and ranged between $\sim$ 1–5.5 $\mathrm{\mu mol\,N\,L^{-1}}$. This represents up to $\sim$ 20% of the initially bioavailable inorganic and organic nitrogen standing stocks. Interestingly, this loss is comparable to the total amount of particulate organic nitrogen that was exported into the sediment

traps at the bottom of the mesocosms in about 20 metres depth. Altogether, this suggests that a significant portion, if not the majority of nitrogen that could be exported to depth, is already lost, i.e. converted to $N_2$ in a relatively shallow layer of the surface ocean, provided oxygen-deficient conditions like during coastal upwelling in our study. Published data for primary productivity and nitrogen loss in all EBUS reinforce such conclusion.

## 1   Introduction

Amongst the most productive marine ecosystems are Eastern Boundary upwelling systems (EBUS), which are mainly fuelled by wind-driven upwelling of dissolved inorganic nutrient-rich deep waters to the surface ocean, stimulating primary and associated higher trophic level productivity (Chavez and Messié, 2009; Kämpf and Chapman, 2016; FAO, 2018). This is particularly true for the Humboldt Current System off Peru (Montecino and Lange, 2009). High productivity and eventual export of organic matter to depth result in marked oxygen consumption by aerobic respiration, leading to so-called sub-surface oxygen-depleted zones as well as virtually anoxic oxygen minimum zones (ODZs and OMZs, respectively) at depth (e.g. Cline and Richards (1972); Paulmier and Ruiz-Pino (2009)). In the absence of oxygen, heterotrophic organic matter decomposition can continue with alternative electron acceptors such as nitrate ($NO_3^-$) or nitrite ($NO_2^-$) via nitric oxide (NO) and nitrous oxide ($N_2O$) to molecular nitrogen ($N_2$), in a series of separate steps carried out by a variety of bacteria (Zumft, 1997). In their entirety these processes are summarised as denitrification. A by-product of denitrification is ammonium ($NH_4^+$) from the remineralised organic matter, but overall denitrification constitutes a net loss of bioavailable nitrogen (Paulmier et al., 2009). A second prevalent nitrogen loss pathway in ODZs and OMZs is autotrophic anaerobic ammonium oxidation (anammox) which utilises $NH_4^+$ and $NO_2^-$ produced by heterotrophic processes to produce energy for carbon dioxide ($CO_2$) fixation and organic matter production (Thamdrup et al., 2006; Brandes et al., 2007).

Observational and modeling studies have estimated the loss of bioavailable nitrogen via water column denitrification and anammox in suboxic ODZs and anoxic OMZs to amount to about 20-35% of the total nitrogen losses ocean-wide (for reviews see Bianchi et al. (2012); Zhang et al. (2020)). In a context of climate/ocean change, which has been projected to enhance the temperature gradient between land and ocean and hence alongshore winds (Di Lorenzo, 2015), upwelling intensity and frequency of deep waters would subsequently increase (Hauri et al., 2013; Wang et al., 2015). Such a scenario has been put forward for the Humboldt Current System off Peru but also other EBUS (Bakun and Weeks, 2008; Varela et al., 2015). Furthermore, due to increasing temperatures the ocean looses oxygen ($O_2$) and OMZs are expanding (e.g. Bopp et al. (2002); Bograd et al. (2008); Stramma et al. (2008); Oschlies et al. (2017)). Together with changes to microbial activity, this modifies biogeochemical properties of upwelled waters including, next to $O_2$, carbonate chemistry speciation, i.e. ongoing ocean acidification further decreases already low deep water pH levels (e.g. Feely et al. (2008); Franco et al. (2018); Schulz et al. (2019)). Changes in upwelling frequency/intensity, oxygen availability, temperature and pH could influence planktonic food web functioning in EBUS, with repercussions for nitrogen loss processes.

To better understand the events following the coastal upwelling of oxygen and nitrogen depleted deep waters, we make use of an off-shore mesocosm setup. This approach allows simulating of upwelling and tracing of biogeochemical element

cycling, as well as associated trophic interactions. We were specifically aiming to address the question of nitrogen cycling, i.e. the build-up and turnover of organic nitrogen pools, their export from the surface to depth, and most importantly, potential loss processes. Because such an approach enables budgeting of the various pools, it will be an alternative and independent assessment of the nitrogen balance in coastal ODZs, next to classical shipboard transects.

## 2 Methods

The current experiment started on February 22, 2017 with the deployment of eight Kiel Off-Shore Mesocosms for Future Ocean Simulations (KOSMOS) about 4 nautical miles (nm) off the Peruvian coast close to Isla San Lorenzo at 12.0555°S and 77.2348°W, and ended on April 16, 2017 after 50 days of sampling. Full details on the experimental setup, and the sampling, manipulation and maintenance schedule can be found in Bach et al. (2020).

### 2.1 Mesocosm setup and sampling

The KOSMOS system comprised 8 mesocosms, consisting of polyurethane bags of 2 m diameter extending to $\sim 18.7$ m depth and the last two metres being a funnel-shaped sediment trap. After two days of thorough flushing and soaking of the bags, which had been pulled under the surface and were open to water exchange through 3 mm mesh on both ends, the bags were secured above the water line and the sediment traps attached, enclosing about 54 $m^3$ of a natural plankton assemblage. Sampling commenced on February 26, i.e. day 1. Sampling days typically started in the morning between 6:30–7:00 by removing the material that had accumulated in the sediment traps with a pump, followed by CTD casts. The CTD (Sea & Sun Technology 90M memory probe) was additionally equipped with an optical oxygen sensor and a calibrated AMT amperometric hydrogen sulphide ($H_2S$) sensor (for details on operation, the other sensors and calibrations/corrections see Schulz and Riebesell (2013)). CTD casts were followed by sampling with an integrating water sampler (IWS, Hyrdobios). Because strong thermal stratification resulted in two distinct water masses at the surface (high oxygen and pH) and at the bottom (low oxygen and pH) of the mesocosms (Fig. 2A, B and C), two separate depth-integrated water samples were taken. According to changes in stratification the depths were adjusted over time (days 1–2: surface 0-5 m and bottom 5-17 m, days 3–28: surface 0-10 m and bottom 10-17 m, and days 30–50: surface 0-12.5 m and bottom 12.5-17 m).

Furthermore, to stabilise and maintain bottom water characteristics (as in the surrounding Pacific), a $NaCl$-brine solution was homogeneously injected below 10 m on day 13 and below 12.5 m on day 33, increasing salinity, and hence stratification, by about 0.7 and 0.5 psu, respectively (see Bach et al. (2020) for details).

While all seawater bulk parameters, such as particulate and dissolved organic matter or dissolved inorganic nutrients were measured on the two depth-integrated samples (see section 2.5 below for details), samples for N-loss process incubations (see section 2.3 below for details) were taken with a Niskin sampler from 15.25 metres depth and treated as gas-sensitive, i.e. filled into two 100 ml glass-stoppered Duran bottles with at least one bottle volume of overflow, closed without headspace and kept cold and in the dark ($\sim 2 - 4$ hours) until processing. A direct comparison of oxygen concentrations in these samples by Winkler-titrations, following the recommendations of Carritt and Carpenter (1966); Bryan et al. (1976); Grasshoff et al.

(1983), with CTD-oxygen-optode measurements revealed an offset by +13 $\mu$mol L$^{-1}$ in the CTD data, although response time hysteresis corrected ($\tau = 1$s) as described in Fiedler et al. (2013). Hence, oxygen concentrations at depth as shown here from CTD casts, are likely to have been significantly lower. The most likely explanation for this offset is that the response time of the sensor was actually slower, on the order of 2–2.5 seconds.

## 2.2 Deep water collection and addition

To simulate upwelling with two distinct OMZ-signatures, in terms of N-deficit, deep water was collected from 90 m at 12.044333°S, 77.377583°W, and 30 m at 12.028323°S, 77.223603°W, on days 5 and 10, respectively. However, both waters had a quite strong N-deficit (N$^*$), in comparison to a typical N/P of 16/1 required for phytoplankton growth (Redfield et al., 1963; Brzezinski, 1985), and will be referred to as 'low N/P' and 'very low N/P' treatments in the following (compare Tab. 1). The deep waters were collected into 100 m$^3$ bags without headspace at the respective depths and sealed once brought back to the surface. Deep water was added by first removing about 20 m$^3$ from each mesocosm and replacing it with the respective deep water that was injected into the bottom layer between 14-17 metres on day 11, and the surface layer between 1-9 metres on day 12. To minimise changes to deep water gas concentrations during injection, water was pumped from two metres depth out of the deep water bags. For further details on collection and addition see Bach et al. (2020).

## 2.3 N-loss processes incubations, measurements and calculations

The two main N-loss processes in oxygen-deficient waters off the Peruvian coast, i.e. denitrification and anammox (Fig. 1) were assayed with incubations of water from 15.25 m depth, using labelled $^{15}$NH$_4$Cl ($\geq$ 98 atom %) or Na$^{15}$NO$_2$ (98 atom %), i.e. by an addition of 3 $\mu$mol L$^{-1}$ each. During incubations with the former, anammox will be traced by the production of $^{29}$N$_2$ ($^{15}$N$^{14}$N), and $^{30}$N$_2$ ($^{15}$N$^{15}$N) if coupled to nitrification (although unlikely as being an oxic process – compare Fig. 1). During incubations with $^{15}$NO$_2^-$, anammox would again produce $^{29}$N$_2$, while denitrification would produce both, $^{29}$N$_2$ and $^{30}$N$_2$ (Fig. 1). Here, a further complication for rate calculations would be coupled nitrate ammonification, aka dissimilatory nitrate or nitrite reduction to ammonium (DNRA), as also leading to $^{30}$N$_2$ production via anammox (compare Fig. 1 and Holtappels et al. (2011)). Incubations were in 12 ml glass Exetainers ('Double Wadded', Labco Ltd.) in duplicates for each of the four time points, i.e. 0, $\sim$ 2, $\sim$ 7 and $\sim$ 20 hours after label addition (in order to avoid potential bottle effects), in the dark and inverted at a fairly constant 17±0.2°C, close to respective in-situ conditions, and similar as described in Dalsgaard et al. (2003), Ward et al. (2009) and Bourbonnais et al. (2021). All sample handling, such as filling of the Exetainers with 8 ml of labelled sample water using an Eppendorf Multipette E3, were carried out in a glove box that had been evacuated three times with a vacuum pump followed by flushing with N$_2$ 4.5 gas (as well as the open Exetainers). To reduce large background $^{28}$N$_2$ levels and facilitate detection of the small isotopic signal of labelled N$_2$ being produced during incubations, the Exetainers were capped and sparged with Helium 5.0 at 3 psi for six minutes on a manifold that could hold all 16 Exetainers of a single mesocosm. Based on previous calculations and measurements, such setup will replace about 20 times the volume of each exetainer (unpublished data). This is lower than the 24 times, reported to ensure that the reduction in O$_2$ concentration is by less than 20% compared to in-situ conditions (Holtappels et al., 2011). A similar reduction would also be observed for

most other gases (Wanninkhof, 1992), except those buffered by conjugate acid-base pairs, such as $H_2S$ or $CO_2$, for which the reduction would be even less. Incubations were stopped at each time point by the injection of 50 $\mu$l of a $ZnCl_2$ solution (50% w/v), followd by thorough mixing.

Exetainer vials were stored upside down at room temperature in the dark. Prior to analysis, the headspace and water was equilibrated at room temperature by placing the Exetainers on a platform mixer at 100 rpm over night. For measurements, 100 $\mu$l of headspace from each Exetainer was injected, using an autosampler (Carvalho and Murray, 2018), into a PLOT GC column at 2 ml/min housed in a Trace GC oven and interfaced with a Thermo Delta V Plus mass spectrometer via a GC Combustion III unit, followed by a liquid nitrogen trap. The latter minimises interference by CO (a constituent in air) and NO (a secondary product during ionisation of water/oxygen, followed by production of reactive oxygen species and recombination with nitrogen) due to imperfect GC column peak separation. The mass spectrometer was calibrated for $N_2$ concentrations by injections of known amounts of air.

Rate calculations were straight forward, as in most $^{15}NH_4^+$ incubations no enrichment over time in $^{29}N_2$ was detected, indicating the absence of anammox. Hence, denitrification in the $^{15}NO_2^-$ incubations (and because $^{30}N_2$ was quite noisy) was calculated from the increase in $^{29}N_2$ and the known ratio of labelled to unlabelled $NO_2^-$ (in the rare cases where anammox was detected in the $^{15}NH_4^+$ incubations, denitrification was corrected for by subtraction). Rates were calculated from the linear regression slopes (Fig. A1) of the increase in the overall amount of N removed (as measured by areas $^{28}A$ and $^{29}A$, and associated $\delta^{15}N$), which was determined by the following conversions (modified from Thamdrup et al. (2006); Holtappels et al. (2011))

$$N_{removal_{t1-3}} \left( \mu mol\, N_2\, L^{-1} \right) = \left[ \left( {}^{15}r\, \frac{\left( {}^{28}A + {}^{29}A \right) cf_{MS}\, \frac{V_h}{V_a}}{f_{N_2\,air}\, cf_{15\,N}} \right)_{t1-3} - \left( {}^{15}r\, \frac{\left( {}^{28}A + {}^{29}A \right) cf_{MS}\, \frac{V_h}{V_a}}{f_{N_2\,air}\, cf_{15\,N}} \right)_{t0} \right] \frac{1000}{V_i} \tag{1}$$

with

$$^{15}r = \frac{{}^{15}R}{1 + {}^{15}R} \tag{2}$$

and

$$^{15}R = \frac{\delta^{15}N}{1000}\, {}^{15}R_{air} + 1 \tag{3}$$

where $^{15}R$ and $^{15}R_{air}$ denote the ratios of $^{15}N/^{14}N$ in a sample gas or air, respectively (the latter determined from measured $^{28}N_2$ and $^{29}N_2$ by Junk and Svec (1958), i.e. 0.00367647, and recommended by Coplen et al. (2002)), $\delta^{15}N$ the resulting isotopic composition (‰), $V_h$, $V_i$ and $V_a$ the volumes of headspace, sample incubated and analysed (ml), respectively, $cf_{MS}$ the determined calibration factor for each measurement run to convert mass spectrometer peak area ($^{28}A + ^{29}A$) into abundance ($\mu$mol), $f_{N_2\,air}$ the fraction of $N_2$ found in equilibrium in the 4 ml of headspace ($V_h$) in relation to the total amount, including the 8 ml of sample water ($V_i$), in an Exetainer, and t1-3 or t0 referring to the respective incubation sampling times. The conversion factor $f_{N_2\,air}$ was calculated from $N_2$ solubility (Hamme and Emerson, 2004) for a lab temperature of 21 °C and a salinity of 35. To extrapolate from measured removal of $^{15}N$ to total N a conversion factor, $cf_{15\,N}$ was calculated that takes into

account the availability of labelled and overall substrate(s), together with the likelihood of $^{29}N_2$ production, i.e. $2F_N(1 - F_N)$ for denitrification incubations (binomial), and $F_N$ for anammox, with $F_N$ denoting the ratio of labelled to total $[NO_2^-]$ or $[NH_4^+]$, respectively.

It is noted that there have been studies which found discrepancies between denitrification calculated using $^{29}N_2$ as above, and $^{30}N_2$, due to non-binomial distributions (De Brabandere et al., 2014; Chang et al., 2014). This has been attributed to so-called intra-cellular '$NO_3^-/NO_2^-$-shunting', which leads to an error in the calculations of labelled to unlabelled substrate, as based on known additions and measured seawater concentrations. As of noisy $^{30}N_2$ data, we cannot check if that was an issue here, yet it would lead to an underestimation of denitrification rates in both cases. Given the good agreement between our rate measurements and a full nitrogen budget (see section 3.2 for details), however, it appears that potential '$NO_3^-/NO_2^-$-shunting'
did not affect our rate measurements significantly.

## 2.4   In-situ substrate limitation of denitrification, total N-loss calculations and orni-eutrophication

In more than half of the denitrification incubations measured N-loss over a period of about 20 hours was higher than the combined concentrations of in-situ $NO_3^-$ and $NO_2^-$. This indicates that the 3 $\mu mol\,L^{-1}$ addition of $Na^{15}NO_2$ alleviated substrate limitation and rates in the incubations were higher than theoretically sustainable at in-situ conditions. Furthermore,
acknowledging that rate measurements at different substrate concentrations in comparison to in-situ conditions are always potential rates, we estimated in-situ rates from a Michaelis-Menten kinetic, by first calculating the maximum rate, $V_{max}$, as

$$V_{max} = \frac{Rate_{meas.}([S]_{in-situ} + [^{15}NO_2^-] + K_{1/2})}{[S]_{in-situ} + [^{15}NO_2^-]} \tag{4}$$

followed by realised rates at in-situ substrate concentrations

$$Rate_{in-situ} = \frac{[S]_{in-situ} V_{max}}{[S]_{in-situ} + K_{1/2}} \tag{5}$$

with $[S]_{in-situ}$ referring to in-situ substrate concentrations of $NO_2^-$ and $NO_3^-$, $[^{15}NO_2^-]$ to the 3 $\mu mol\,L^{-1}$ label addition, and $K_{1/2}$ to the reaction half-saturation constant. As we have not performed any substrate vs. rate essays ourselves and since there is only little information on the kinetics of water column denitrification, we adopted a conservative approach and chose the highest published $K_{1/2}$ of 5 $\mu mol\,L^{-1}$ (Michiels et al., 2019), over the 2.9 and 2.5 $\mu mol\,L^{-1}$ of Jensen et al. (2009) and Dalsgaard et al. (2013), respectively.

Total N-loss by denitrification and anammox was calculated only for the first 38 days of the experiment. The reason was to keep the estimates by rate measurements comparable to estimates from a full N-budget of the mesocosms, as after day 40 massive perturbations of the latter became obvious. This was caused by birds aggregating on the roofs of the mesocosms and depositing nitrogen-rich faeces (orni-eutrophication). The onset of this perturbation was estimated by the sudden increase in total particulate phosphorus deposition rates in the sediment traps, most likely from excrements, as phosphate concentrations
during this time were relatively constant (Fig. A2 – for details see Bach et al. (2020)). Total N-loss was then estimated by

summing up hourly in-situ rate estimates (Eq. 5) for each mesocosm (in case also anammox was detected the measured rate was added to that of denitrification), factoring in the varying measurement intervals and thus filling in days without measurements. This hourly $N_2$ loss was then multiplied by 24 hrs per day $\times 2$ (conversion between $N_2$ to N) and divided by 3 (average contribution of bottom layer water to overall mesocosm volume), converting it to total N-loss over the first 38 days of the experiment.

## 2.5 Ancillary measurement parameters

Dissolved inorganic nutrient concentrations in seawater, i.e. $NO_3^-$ $NO_2^-$ $NH_4^+$ (together referred to as DIN), $PO_4^{3-}$ and $Si(OH)_4$ were measured colorimetrically on a segmented flow analyser (QuAAtro, SEAL Analytical) on site (for details see Bach et al. (2020)).

Dissolved organic nitrogen (DON) was calculated from a mass balance by subtracting measured DIN from total dissolved nitrogen (TDN) concentrations. The latter was also determined by segmented flow analysis after an oxidising step with Oxisolv (Merk) for 30 minutes in an autoclave.

Particulate organic nitrogen and carbon (PON, POC) concentrations in the water column were determined by filtering known amounts of seawater over pre-combusted GF/F filters that were stored frozen until analysis on an elemental analyser (EuroVector EA 3000 - for details see Bach et al. (2020)).

The removal of nitrogen from the water column via sedimentation ($N_{sed}$) was determined on material collected every other day from the sediment traps which was quantitatively precipitated, freeze-dried, weighed and an aliquot measured on an elemental analyser (EuroVector EA 3000 - for details see again Bach et al. (2020) and Boxhammer et al. (2016)).

## 2.6 Statistical analyses

In order to assess what drives denitrification rates, stepwise multiple linear regressions with interaction terms (MLRs) were carried out. Prior to regressions, outliers in estimated in-situ denitrification rates (see section 2.4 and Tab. 2 for details) were identified in a box-whisker plot and removed, i.e. five values with rates higher than $10\,\mathrm{nmol\,L^{-1}h^{-1}}$. In order to avoid overfitting and find a balance between model complexity and explanatory power, we followed a backward elimination process, starting with the full seven potential measured predictors, i.e. PON, $PON_{sed}$, DON, $NO_3^-$, $NO_2^-$, $O_2$ and $H_2S$, all measured in the bottom layer of the mesocosms (with the exception of sedimenting $PON_{sed}$). Note that we have opted to not include POC, as being highly co-correlated with PON, as well as DOC which was not measured in the bottom layer (but probably would have a similar issue). In a next step, MLRs with all possible combinations of 6 potential predictors, out of the overall 7, and their resulting $R^2$ were calculated, which was followed by further MLRs, subsequently reducing the number of predictors each time by 1 (a total of 119 MLRs were fitted). Calculations were performed using the functions 'boxplot', 'stepwisefit' and 'plotEffects' in MATLAB.

## 3    Results

The experiment took place during the 2017 coastal El Niño, which was characterised by three significant surface ocean warming events throughout January to April (Garreaud, 2018) with the last two, end of February and mid March, clearly evident by water surface temperatures above 22°C at our mooring site (Fig. 2A). The El Niño was accompanied by torrential rains further inland which was reflected by periods of significant reductions in surface ocean salinity (Fig. 2B), coinciding with water discharge more than twice the typical rates of the nearby river Rimac (Fig. A3). During our experiments there were, however, also periods of deep water upwelling, as evidenced by colder surface ocean temperatures, as well as reduced oxygen saturation states and pH levels, reaching down to ~30% and 7.5 (total scale), respectively (Fig. 2C, D).

### 3.1    Temporal changes in oxygen, inorganic and organic nitrogen as well as hydrogen sulphide in the bottom layer of the mesocosms

Thermal stratification of Pacific Ocean and mesocosm waters during the initial phase of the experiment meant strict separation of well oxygenated surface (~0-10 metres) from oxygen-depleted bottom waters (Fig. 2C). And while in the surrounding Pacific bottom waters remained oxygen depleted throughout, corresponding oxygen levels in the mesocosms started to increase (Fig. 2 and 3E). This was caused by cooling of the surface waters due to upwelling in the surrounding Pacific and resulting mixing of surface and bottom waters in the mesocosms. Such artificial behaviour was mitigated by increasing bottom layer salinity, and therefore stratification, on days 13 and 33, which brought oxygen concentrations at depth down to typical Pacific levels again (Fig. 3E).

All mesocosms started with significant amounts of dissolved inorganic nitrogen present as $NO_3^-$ and $NO_2^-$, which however were quickly depleted within the first two weeks (Fig. 3C, D). The deep water addition significantly increased $NO_2^-$ concentrations in the 'low N/P' treatments, although it made only a minor contribution to the overall nitrogen budget (see following sections for details).

Initial particulate and dissolved organic nitrogen concentrations (PON, DON) at depth were similar in all mesocosms (between $6-10\,\mu\mathrm{mol\,L^{-1}}$), and while there was no clear temporal trend for PON, DON saw a steady decline by about 30% until day 40 (Fig. 3A, B).

$H_2S$ concentrations at depth were in the micro-molar range in all mesocosms as well as the surrounding Pacific and mostly oscillated between $3-10\,\mu\mathrm{mol\,L^{-1}}$ (Fig. 3F), equivalent to 0.1–0.3 ppm. In contrast concentrations in surface waters were mostly in the high to low nano-molar range (data not shown).

### 3.2    Rates of denitrification and anammox, and overall nitrogen loss

Measured denitrification rates in the 24 hour incubations were similar in all mesocosms, ranging between less than 1 to up to $\sim80\,\mathrm{nmol\,L^{-1}h^{-1}}$ (Tab. A1). More importantly however, towards the end, measured rates exceeded those theoretically sustainable by substrate availability, i.e. combined in-situ $NO_3^-$ and $NO_2^-$ concentrations. In comparison, measured denitrifica-

tion rates in samples from the surrounding Pacific were comparable, although more variable between consecutive measurement days.

Estimates for in-situ denitrification (Eq. 5) were similar to measured rates at the beginning of the experiment, when substrate availability was higher than or close to the half-saturation constant for denitrification. Towards the end, at high to low nano-molar substrate concentrations, in-situ estimates were significantly lower (compare Tab. 2 and Tab. A1).

Anammox was only detected on day 12 in mesocosms 7 and 8, at rates significantly smaller than those typical for denitrification (Tab. 2). In contrast, anammox was occasionally detected in the surrounding Pacific Ocean, i.e. on days 8, 12 and 46.

When comparing potential nitrogen loss, calculated as the sum of in-situ estimates of denitrification and anammox, over the first 38 days prior to the onset of orni-eutrophication (see section 2.4 and Tab. 2 for details), with estimates of total nitrogen losses in each mesocosm from a nitrogen budget approach (compare Tab. 2 and Fig. 4 and see next section for details), they were similar, with a mean of $2.69 \pm 1.18\,\mu\mathrm{mol\,L^{-1}}$ and $3.64 \pm 2.12\,\mu\mathrm{mol\,L^{-1}}$, respectively. It is acknowledged, however, that there was no statistically significant correlation between the two approaches.

## 3.3 Nitrogen budget

Summing up all organic and inorganic nitrogen species (excluding dissolved gases) in the water column, i.e. PON, DON, $NO_3^-$, $NO_2^-$, $NH_4^+$, $N_{\mathrm{sed}}$ (nitrogen exported through the sediment traps), and $N_{\mathrm{DW}}$ (overall deep water addition changes to the various nitrogen pools), revealed a first phase in all mesocosms for which the budget did not seem to be closed, as evidenced by a constant decline in total N during the first two weeks (Fig. 4). This was followed by a steady increase in total N until day 40, which was still below starting levels in all mesocosms. This net loss reflects, as we will argue below, the combined nitrogen loss processes such as denitrification and anammox, exceeding nitrogen fixation, which was less than $0.1\,\mu\mathrm{mol\,L^{-1}}$ over the entire water column in this period (Kittu pers. comm.). Finally, the last ten days were characterised by a rapid increase in total N, mostly driven by rising surface PON concentrations, following continuous biomass production fuelled by external orni-eutrophication, i.e. the introduction of nitrogen-rich bird faeces to the mesocosms (see Bach et al. (2020) for details).

## 3.4 Stepwise multiple linear regressions

In terms of $R^2$ there was a turning point in the number of potential main variables to explain in-situ denitrification rates in stepwise multiple linear regressions (MLRs) with interactions. Above 4–5 independent variables, $R^2$ only increased slightly, while below it decreased relatively quickly (Fig. 5A). Both models with the highest $R^2$, for 4 and 5 main variables, identified $NO_2^-$, $NO_3^-$ and PON as those with the highest positive main effect sizes, with the remaining variables having relatively low effect sizes (Fig. 5B, D, Tab. 3). Finally, $O_2$ had a small negative main effect size in both models.

The resulting linear regression of in-situ (compare Tab. 2) versus predicted denitrification rates slightly deviated from an ideal 1:1 relationship and had a small, yet positive y-axis intercept, meaning that lower denitrification rates would be over-while higher ones underestimated (Fig. 5C).

## 4 Discussion

A discussion on the unusual situation of a coastal El Niño during our study period, i.e. significant warming of surface waters, increasing stratification and reducing upwelling intensity/frequency can be found elsewhere (Bach et al., 2020). Most importantly however, there were also multiple upwelling events in the surrounding Pacific during this time which were of similar magnitude to our experimental upwelling (Chen et al., 2021), providing the natural context.

### 4.1 Denitrification rates and nitrogen budgets

Denitrification rates measured in the incubations of water from the mesocosms and the surrounding Pacific of up to $\sim$ 80 $\mathrm{nmol\,N_2\,L^{-1}\,h^{-1}}$ (although mostly well below at a median of 12.4) were within the range of reported rates in Peruvian ODZ and OMZ waters (Farías et al., 2009; Dalsgaard et al., 2012). This was despite significantly higher oxygen concentrations here than typical for suboxic ODZs, i.e. less than $\sim$6 $\mu\mathrm{mol\,L^{-1}}$ (Tyson and Pearson, 1991; Yang et al., 2017). Although, denitrification can be relatively insensitive to oxygen concentrations of up to at least 30-40 $\mu\mathrm{mol\,L^{-1}}$ (Farías et al., 2009), which is higher than levels in our study for most of the time (see Fig. 3 but also section 4.2.2).

Most interestingly, measured denitrification rates in incubations of mesocosm waters during the second half of the experiment exceeded those theoretically sustainable at in-situ substrate availability (compare section 2.4 and Tab. 2). Nitrification, supplying additional nitrite for denitrification, can operate at low micro-molar (and even nano-molar) levels such as those found in our study (Bristow et al., 2016). However, a significant contribution to measured denitrification rates is unlikely as there and elsewhere (Peng et al., 2016; Santoro et al., 2021) nitrification rates were usually at least an order of magnitude lower than our measured denitrification rates. Hence, the reason for higher measured rates is most likely found in the fact that both $\mathrm{NO_3^-}$ and $\mathrm{NO_2^-}$ concentrations had dropped below 0.1 or even 0.01 $\mu\mathrm{mol\,L^{-1}}$ during this stage, indicating that the denitrifying community was actually substrate limited. Such observation was also made by Michiels et al. (2019), although in a temperate Fjord system. In comparison, denitrification rates in the surrounding Pacific waters had a higher day to day variability and substrate limitation only occurred once. This is to be expected in a much more variable system, characterised by frequent upwelling events of oxygen depleted waters (Fig. 2) with relatively higher $\mathrm{NO_3^-}$ and $\mathrm{NO_2^-}$ concentrations (Fig. 3C, D).

Calculated Michaelis-Menten kinetic scaled estimates for in-situ denitrification in the mesocosms and the surrounding Pacific were lower than measured rates during the incubations. This was particularly the case during the second half of the experiment, which reinforces the notion of substrate limitation.

The primary drivers of in-situ denitrification rates appeared to be the availability of $\mathrm{NO_2^-}$ and $\mathrm{NO_3^-}$, followed by particulate organic matter (nitrogen), as indicated by multiple linear regression and effect size analysis (Fig. 5). This suggests that heterotrophic rather than chemolithotrophic denitrification was dominant (compare section 4.2.2), as all three are substrates for the former process and eventually limit rates. The reason for $\mathrm{NO_2^-}$ rather than $\mathrm{NO_3^-}$ concentrations explaining rates of denitrification in one of the MLRs could be found in the following: As denitrification from $\mathrm{NO_3^-}$ to $\mathrm{N_2}$ involves multiple and independent steps and organisms, the correlation between $\mathrm{N_2}$ production rate and a substrate concentration should become better the closer one gets to the end of this chain (Fig. 1). For example, there should be a perfect correlation between $\mathrm{N_2O}$

concentrations and $N_2$ production rate, and $NO_3^-$ concentrations and their turn-over to $NO_2^-$ are meaningless if the intermediate steps to nitric and nitrous oxide are blocked or constitute a bottle-neck. This also would contribute to the finding that denitrification rate measurements based on $^{15}NO_3^-$ can be lower than those based on $^{15}NO_2^-$ (Hamersley et al., 2007).

Overall nitrogen loss, calculated from in-situ denitrification rates over the first 38 days of the experiment prior to the onset of orni-eutrophication, was comparable to an alternative estimate that was based on a full nitrogen budget in each mesocosm (Fig. 4, Tab. 2). The means of both methods, i.e. $2.69 \pm 1.18$ and $3.64 \pm 2.12$, were insignificantly different, although there was no statistically significant correlation when all mesocosms were compared. However, given that the nitrogen budget calculations involved a mass balance of seven entities with individual measurement uncertainties, this probably does not come as a surprise.

There were also no statistically significant differences between the two deep water addition treatments, which was most likely due to relatively small differences in dissolved inorganic nitrogen in both waters in relation to the overall nitrogen pool in the mesocosms, as well as a similar N-deficit in both deep water batches (Tab. 1). However, N-loss estimated from in-situ denitrification rates in the low N/P mesocosms was significantly lower than the budget estimate. It is noted, however, that in-situ denitrification rates are a conservative estimate, potentially underestimating true loss. For instance, adopting a lower

half-saturation rate constant of $2\ \mu mol\,L^{-1}$ (Eqs. 4 and 5) would remove any statistically significant difference between the two approaches. Furthermore, $NO_3^-$ and $NO_2^-$ standing stocks most likely underestimate availability, and hence in-situ denitrification rates calculated here, as of hidden turn-over.

Finally, modeling exercises suggest that nitrogen loss in the water column is linked with mature El Niño/La Niña periods, with up to 70% reduced rates during the former, most likely linked to increased water column oxygen concentrations at

320 reduced upwelling/enhanced stratification (Deutsch et al., 2011; Yang et al., 2017), as well as increasing mesoscale turbulence and associated offshore nutrient export by eddies (Espinoza-Morriberón et al., 2017). This suggests that the rates observed here could actually be significantly higher in the more frequent La Niña periods.

## 4.2 Rates of anammox and the lack thereof in the mesocosms

Anammox and denitrification were first reported in the Eastern Tropical South Pacific (ETSP) in 2006 (Thamdrup et al., 2006),

and the ETSP remains the most frequently sampled and most thoroughly characterised open ocean OMZ region. Anammox was the dominant N loss process in several studies, where denitrification was either not (Hamersley et al., 2007) or only sporadically (Kalvelage et al., 2013) detected. Subsequent studies in the ETSP detected both processes, and anammox was a substantial and mostly dominant part of the total N loss: average 78% anammox along a Chilean coastal transect (Dalsgaard et al., 2012), 82–90% at two high resolution stations (De Brabandere et al., 2014), and $49 \pm 20$% over four high resolution stations (Babbin et al.,

2020).

Anammox requires a source of $NH_4^+$ and $NO_2^-$, which must either be produced in situ by remineralisation of organic matter in the absence of oxygen (as for instance by denitrification) or must be transported into the system from elsewhere, e.g., from adjacent sediments (Ward, 2013). When both processes are supported by organic matter remineralisation, there is theoretical (Paulmier et al., 2009; Koeve and Kähler, 2010) and experimental evidence (Babbin et al., 2014) that the ratio

of denitrification to anammox should be connected to the elemental composition of organic matter being decomposed, at

least on larger spatial and temporal scales. The reasoning behind this connection is that complete anaerobic organic matter decomposition by denitrification produces $NH_4^+$ and $N_2$ in quantities mostly dependent on the carbon to nitrogen ratio (C/N) of the organic matter being decomposed. Hence, this ratio dictates the denitrification (only utilising $NO_3^-$ and or $NO_2^-$) to anammox (utilising both $NO_2^-$ and $NH_4^+$) ratio in steady state (Babbin et al., 2014). However, DNRA could also supply $NH_4^+$

for anammox. Although there have been reports of DNRA in the ETSP (Lam et al., 2009), it is usually negligible in the water column (De Brabandere et al., 2014; Kalvelage et al., 2013) and rather restricted to shallow coastal systems dominated by sediments (Jensen et al., 2011). In the absence of DNRA, complete anaerobic decomposition of average phytoplankton-derived organic matter, i.e. $C_{106}H_{175}O_{42}N_{16}P$ (Anderson, 1995), would require a $\sim 28\%$ contribution of anammox to overall N-loss via these two processes (Babbin et al., 2014).

The reason for an anammox dominance in several studies mentioned above, despite above outlined stoichiometric constraints could be partial denitrification of $NO_3^-$ to $NO_2^-$ without the following steps leading to $N_2$ production, supplying both $NO_2^-$ and $NH_4^+$ for anammox (Lam et al., 2009; Jensen et al., 2011; Kalvelage et al., 2013; Peters et al., 2016). Furthermore, the ratio of $NO_2^-$ to $NH_4^+$ produced by organic matter decomposition of a certain C/N ratio could also be influenced by anaerobic chemolithotrophic nitrite oxidation (compare Fig. 1), looping $NO_2^-$ back to $NO_3^-$ for further partial denitrification and

associated $NH_4^+$ production (Babbin et al., 2020).

In summary, anammox in the absence of denitrification is difficult to explain (although partial denitrification is not easily detected), while denitrification in the absences of anammox is rarely observed in the ETSP but presents no stoichiometric conflicts.

### 4.2.1 Organic matter C/N in mesocosms

For the first 38 days of the experiment, particulate organic matter in the bottom layer of the mesocosms, where the N-loss process incubation samples were taken from, had a C/N of $\sim 7$-8 (compare Bach et al. (2020)), which would not change the theoretical anammox contribution of $\sim 28\%$ by much. However, this ratio could be misleading as the particulate organic matter will be comprised by more and less labile fractions. A good proxy for that should be to just consider the freshly produced organic matter, i.e. in our case the particulate and dissolved matter in the sun-lit surface layer of the mesocosms. Interestingly,

just looking at C/N of standing stocks for particulate organic matter, it was initially lower than in the bottom layer, indicative of preferential nitrogen remineralisation at depth, but increased to values of $\sim 10$ in all mesocosms but one (no bloom of the dinoflagellate *Akashiwo sanguinea* - see Bach et al. (2020) for details) after the deep water addition (Fig. A4). Similarly, C/N in dissolved organic matter started already at about 10, increasing to up to 30 after the deep water addition, and when summed up to total organic matter C/N levels of up to 20 were reached (Fig. A4). Increasing C/N from 6.625 to 20 would

reduce the theoretical anammox contribution to only about 10%. Furthermore, as mentioned already above, it should not be the standing stocks that are considered but the actual fresh production. Unfortunately, neither C nor N production were directly measured, but gauging the change in total organic carbon (TOC) concentrations after deep water addition suggests that at least $\sim 100\ \mu mol\,L^{-1}$ were produced (Fig. A4). In contrast, total organic nitrogen (TON) concentrations hardly changed at all, suggesting even higher C/N in freshly produced organic matter than in measured standing stocks. This could be the result of

general dissolved inorganic nitrogen limitation and hence carbon over-consumption (Toggweiler, 1993), but also changes in phytoplankton community composition (see Bach et al. (2020) for details). It is noted though that changes in organic matter standing stocks over time are not necessarily a good indicator of the quantity and quality of fresh production due to unknown and potentially different turnover times for each element, although probably better than using standing stocks themselves. In this sense the ODZ/OMZ signature of upwelled water, i.e. its nitrogen deficit, should influence the ratio of denitrification to anammox that is subsequently measured. Hence, regions with one or the other process dominating could be indicative of the upwelling source water history.

### 4.2.2 Oxygen and hydrogen sulphide in mesocosms

Varying ratios of anammox to denitrification could also be the result of different sensitivities of both processes to prevailing oxygen concentrations. However, while there are a few experiments that have directly addressed oxygen sensitivity of various N-cycling processes there appears to be no straight-forward answer. For instance, in manipulative experiments Jensen et al. (2008) found anammox at oxygen concentrations of up to $\sim 12.5\,\mu\mathrm{mol\,L^{-1}}$, with denitrification not encountered at all. This is similar to Kalvelage et al. (2011) who observed anammox to cease above $\sim 20\,\mu\mathrm{mol\,L^{-1}}$. And although no full denitrification was observed, the first step from $NO_3^-$ to $NO_2^-$ was found to occur even at the highest oxygen concentration of $\sim 25\,\mu\mathrm{mol\,L^{-1}}$. This is in turn consistent with nitrate reduction observed to at least $N_2O$ at oxygen concentrations of up to $\sim 30\text{-}40\,\mu\mathrm{mol\,L^{-1}}$ (Frey et al., 2020). In contrast, both anammox and denitrification can already be significantly/completely inhibited by oxygen additions of 3 or $8\,\mu\mathrm{mol\,L^{-1}}$ (Babbin et al., 2014) or even lower (Dalsgaard et al., 2014). In essence, there is no clear indication of one process being more sensitive to oxygen than the other, potentially also related to variability of oxygen concentrations on small scales, i.e. in mircoenvironments around particles which are not captured by bulk seawater oxygen concentration measurements. And indeed there is evidence that microbial diversity in the OMZ is critically linked to particles, as are rates of denitrification and anammox (Ganesh et al., 2014, 2015).

The situation appears to be similar for $H_2S$, which furthermore has been observed to reach $10\,\mu\mathrm{mol\,L^{-1}}$ and even higher in the coastal sub-surface ETSP close to our study location (Callbeck et al., 2018, 2019). $H_2S$ has been reported to completely inhibit anammox at $3\,\mu\mathrm{mol\,L^{-1}}$ (Jensen et al., 2008), a concentration similar to that in the bottom layer of our mesocosms, while Dalsgaard et al. (2014) found no effect on $N_2$ production by anammox, nor denitrification, at slightly lower levels of $1\,\mu\mathrm{mol\,L^{-1}}$. Concerning denitrification, Dalsgaard et al. (2013) reported no effects of $H_2S$ at even higher concentrations of up to $10\,\mu\mathrm{mol\,L^{-1}}$, which is in the range of maximum concentrations found in our study, as well as stimulation in some instances, although most likely related to chemolithotrophic as opposed to heterotrophic denitrification (see Dalsgaard et al. (2013); Bonaglia et al. (2016) and references therein). Chemolithotrophic denitrification coupled to $H_2S$ oxidation has also been hypothesised for coastal high $H_2S$ stations off Peru by Kalvelage et al. (2013). Interestingly, by day 16 and onwards the anammox functional marker gene *hzo* (Schmid et al., 2008) was not anymore detectable in any of the mesocosms (data not shown), which could be linked to relatively high $H_2S$ concentrations, explaining the lack of anammox, at least for most parts of the experiment.

## 5  Conclusions: ODZ/OMZ nitrogen budget implications

The loss of on average 3–4 $\mu\mathrm{mol\,L^{-1}}$ of nitrogen in our mesocosms fits well to zonal estimates by DeVries et al. (2012) for the ODZ in the Eastern South Pacific between 75–100 metres depth where oxygen levels were similar to those encountered here (Chang et al., 2010), although the latter study reported a 2–3 times higher nitrogen deficit. Furthermore, the overall amount of nitrogen loss measured in the Pacific was at the upper end of ranges encountered in the mesocosms, potentially connected to substrate limitation in the mesocosms during the second half of the experiment (Tab. 2).

In contrast to shipboard measurements, the mesocosms offer the unique opportunity to put the various nitrogen pathways, i.e. loss as opposed to initial bioavailable standing stocks and export, into context. Between $\sim$ 1–5.5 $\mu\mathrm{mol\,L^{-1}}$ of nitrogen were lost as $N_2$, representing up to 20% of the initially bioavailable inorganic and organic nitrogen until day 38 of the experiment (compare Tab. 2 and Fig. 4). Interestingly, the amount of particulate organic nitrogen being exported below 20 metres, i.e. the approximate depth of the sediment traps at the bottom of the mesocosms (compare Fig. 1 in Bach et al. (2020)), was in the same range. This indicates that in the Peruvian EBUS about half of the nitrogen that could be exported to depth would already be lost, i.e. converted to $N_2$, in a relatively shallow layer of the surface ocean, provided oxygen deficient conditions during coastal upwelling as in our study (Fig. 2). Furthermore, over the entire pelagic water column, nitrogen loss of exported organic matter is likely to be even higher, suggesting that the majority of the dissolved inorganic nitrogen assimilated during new production (equalling export production on larger scales) should actually be lost in EBUS.

Similar conclusions can also be reached by alternative means, i.e by starting with global export production. Recent estimates based on observations and models range between $\sim$5–15 $\mathrm{Pg\,C\,y^{-1}}$, including next to the gravitation driven biological pump also those by particle injection (see Boyd et al. (2019) and refs. therein). Assuming a Redfieldian molar C/N of 6.625 (Redfield et al., 1963) would translate to 0.75–2.26 $\mathrm{Pg\,N\,y^{-1}}$ being exported. Furthermore, considering that about 5% of global primary production, and hence potential export production, is located in the surface ocean of the four major EBUS (Carr, 2002), Arabian Sea (Vijayaraghavan and Krishna Kumari, 1989) and Bay of Bengal (Poornima et al., 2020) above the ODZs and OMZs, about 38–113 $\mathrm{Tg\,N\,y^{-1}}$ could potentially be lost. In comparison, estimates of water column denitrification/anammox, i.e. 20–35% of total marine losses of about 260 $\pm$ 100 $\mathrm{Tg\,N\,y^{-1}}$ (see Zhang et al. (2020) and references therein), then range between 52 $\pm$ 20 and 91 $\pm$ 35 $\mathrm{Tg\,N\,y^{-1}}$, indicating that a significant portion, if not the majority, of the exported nitrogen is indeed lost in ODZs and OMZs.

Nitrogen cycling in ODZs and OMZs currently plays a very important role in the overall marine nitrogen budget. However, the magnitude and direction of change in the actual nitrogen loss term in response to ongoing climate and ocean change (e.g. ocean stratification, acidification and/or changes in temperature and oxygen levels) is uncertain. This issue is further complicated by uncertainties in future primary productivity and organic matter export estimates. For instance, depending on the representative concentration pathway, future export production could decrease as a result of changes to community structure (see Bindoff et al. (2019) for details and refs. therein). In summary, future changes in upwelling intensity and frequency, as well as the other potential biotic and abiotic factors mentioned above, could change the nitrogen (im)balance in ODZs and OMZs, having a significant impact on the overall marine nitrogen budget.

*Data availability.* The majority of data is available at https://doi.org/10.1594/PANGAEA.923395 (Bach et al., 2020), with some additional being presented here in tables.

*Author contributions.* KGS, LB, TB, AL, and UR designed the experiment. KGS, JA, LB, TB, MI, VK, AL, JaM, JuM, FM, EvdE and UR contributed to sampling and various measurements. DE helped with $^{15}$N data interpretation and BBW introduced KGS to denitrification and anammox assay incubations. All authors analysed the data and KGS wrote the manuscript with co-authors commenting.

*Competing interests.* The authors declare that they have no conflict of interests.

*Acknowledgements.* This project was supported by the Collaborative Research Centre SFB 754 Climate- Biogeochemistry Interactions in the Tropical Ocean financed by the German Research Foundation (DFG). Additional funding was provided by the EU project AQUACOSM and the Leibniz Award 2012 granted to UR. We thank all participants of the KOSMOS-Peru 2017 study for assisting in mesocosm sampling and maintenance. We are particularly thankful to the staff of IMARPE for their support during the planning, preparation and execution of this study and to the captains and crews of BAP MORALES, IMARPE VI and BIC HUMBOLDT for support during deployment and recovery of the mesocosms and various operations during the course of this investigation. Special thanks go to the Marina de Guerra del Peru, in particular the submarine section of the Navy of Callao, and to the Dirección General de Capitanías y Guardacostas. We also acknowledge strong support for sampling and mesocosm maintenance by Jean-Pierre Bednar, Gabriela Chavez, Susanne Feiersinger, Peter Fritsche, Paul Stange, Anna Schukat, Michael Krudewig. We want to thank Club Náutico Del Centro Naval for excellent hosting of our temporary filtration laboratory, office space and their great support and improvisation skills after two of our boats were lost. This work is a contribution in the framework of the Cooperation agreement between the IMARPE and GEOMAR through the German Ministry for Education and Research (BMBF) project ASLAEL 12-016 and the national project Integrated Study of the Upwelling System off Peru developed by the Direction of Oceanography and Climate Change of IMARPE, PPR 137 CONCYTEC. Furthermore, the SERVICIO NACIONAL DE METEOROLOGÍA E HIDROLOGIA DEL PERU - SENAMHI - is thanked for providing the Rimac discharge data. Finally, Matheus Carvalho de Carvalho is thanked for $N_2$ isotope measurements.

*Financial support.* This research has been supported by the Collaborative Research Center SFB 754 Climate-Biogeochemistry Interactions in the Tropical Ocean financed by the German Research Foundation (DFG), the Leibniz Award 2012 (granted to Ulf Riebesell), the Helmholtz International Fellow Award 2015 (granted to Javier Arístegui) and EU funding by AQUACOSM granted to a number of participants, including Javier Arístegui and Kai G. Schulz.

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

**Table 1.** Concentrations of dissolved inorganic and organic nitrogen and phosphorus ($\mu\text{mol}\,\text{L}^{-1}$), as well as silicate in the two deep water batches and associated inorganic N/P/Si in comparison to a Redfieldian ratio of 16/1/16 (see text for details), as well as the inorganic N-deficit calculated as $N^* = DIN - 16 \times PO_4^{3-}$, with DIN denoting the sum of $NO_3^-$ $NO_2^-$ and $NH_4^+$.

| | $NO_3^-$ | $NO_2^-$ | $NH_4^+$ | DON | | $PO_4^{3-}$ | DOP | | $Si(OH)_4$ | | N/P/Si | | $N^*$ |
|---|---|---|---|---|---|---|---|---|---|---|---|---|---|
| **Low N/P DW** | 1.1 | 2.9 | 0.3 | 5.5 | | 2.5 | 0.2 | | 19.6 | | 1.7 / 1 / 7.8 | | -35.7 |
| **Very low N/P DW** | 0 | 0 | 0.3 | 5.2 | | 2.6 | 0.2 | | 17.4 | | 0.1 / 1 / 6.7 | | -41.3 |

**Table 2.** Calculated in-situ denitrification rates (nmol $N_2$ $L^{-1}$ $h^{-1}$) in the low N/P and very low N/P deep water addition mesocosms and the surrounding Pacific (see Methods section 2.4 for details), together with those for *anammox** (nmol $N_2$ $L^{-1}$ $h^{-1}$), when encountered, at various days. N-loss refers to the total amount of nitrogen (μmol N $L^{-1}$) being lost through these processes for the first 38 days, for which an N-budget can be calculated (compare Fig. 4). N-loss was calculated for each mesocosm until day 38 by taking into account the varying measurement intervals and assuming an average contribution of bottom water to the overall mesocosm volume of one third. For details on calculations see Methods.

| Mesocosm | In-situ denitrification rates and/or *anammox** (nmol $N_2$ $L^{-1}$ $h^{-1}$) | | | | | | | | | | N-loss (μmol N $L^{-1}$) T1-38 | N-budget (μmol N $L^{-1}$) T1-38 |
|---|---|---|---|---|---|---|---|---|---|---|---|---|
| **low N/P** | T8 | T12 | T16 | T22 | T26 | T30 | T34 | T38 | T42 | T46 | | |
| **M2** | 12.45 | 6.33 | 1.35 | 1.21 | 0.65 | 1.50 | 0.44 | 0.40 | 0.52 | 0.20 | **-2.39** | **-5.55** |
| **M3** | 4.29 | 9.52 | 7.90 | 6.62 | 2.06 | 0.57 | 0.74 | 5.79 | 1.67 | 0.21 | **-2.89** | **-3.38** |
| **M6** | 3.26 | 17.01 | 2.16 | 2.75 | 0.59 | 0.35 | 0.26 | 0.26 | 0.16 | 0.06 | **-2.00** | **-4.46** |
| **M7** | 4.89 | 3.62 | 2.43 | 0.23 | 0.48 | 0.38 | 0.00 | 0.17 | 0.17 | 0.15 | **-1.10** | **-3.00** |
| | | *0.44** | | | | | | | | | | |
| **Treatment Mean±SD** | | | | | | | | | | | **-2.10±0.76** | **-4.10±1.15** |
| **very low N/P** | | | | | | | | | | | | |
| **M1** | 6.04 | 14.46 | 7.77 | 1.67 | 0.50 | 1.55 | 3.80 | 0.03 | 0.32 | 0.52 | **-2.73** | **-3.44** |
| **M4** | 0.66 | 32.71 | 7.26 | 4.81 | 7.58 | 4.19 | 0.10 | 0.08 | 0.04 | 0.01 | **-3.87** | **-7.23** |
| **M5** | 4.67 | 60.92 | 2.11 | 0.73 | 0.51 | 0.18 | 0.44 | 0.23 | 0.22 | 0.07 | **-4.79** | **-1.20** |
| **M8** | 9.13 | 5.77 | 1.88 | 0.34 | 0.25 | 0.45 | 0.23 | 0.15 | 0.15 | 0.02 | **-1.76** | **-0.87** |
| | | *0.46** | | | | | | | | | | |
| **Treatment Mean±SD** | | | | | | | | | | | **-3.29±1.32** | **-3.19±2.93** |
| **Overall Mean±SD** | | | | | | | | | | | **-2.69±1.18** | **-3.64±2.12** |
| **PACIFIC** | 8.32 | 1.96 | 5.94 | 17.27 | 0.09 | 2.67 | 0.21 | 0.00 | 0.25 | 48.46 | **-4.47** | |
| | *7.48** | *0.16** | | | | | | *1.30** | | *1.12** | | |

**Table 3.** Multiple linear regression statistics, describing denitrification in response to various environmental variables, for the best 5 variable model (in terms of $R^2$) with interactions (compare Fig. 5C), as well as the best 4 variable model.

| | Estimate | SE | t | p | | | Estimate | SE | t | p |
|---|---|---|---|---|---|---|---|---|---|---|
| (Intercept) | 2.5529 | 1.5437 | 1.6537 | 0.104 | | (Intercept) | 0.7507 | 1.0888 | 0.6894 | 0.4933 |
| PON | 0.0039 | 0.1037 | 0.0377 | 0.970 | | PON | -0.0382 | 0.1118 | -0.3420 | 0.734 |
| $NO_3^-$ | 3.2562 | 1.1318 | 2.8770 | 0.006 | | $NO_3^-$ | 5.9709 | 0.7551 | 7.9072 | <0.001 |
| $NO_2^-$ | -5.1120 | 9.3374 | -0.5475 | 0.586 | | $NO_2^-$ | -29.494 | 4.8048 | -6.1385 | <0.001 |
| $O_2$ | -0.0964 | 0.0515 | -1.8742 | 0.066 | | $O_2$ | $4.72 \times 10^{-4}$ | 0.0234 | 0.0202 | 0.984 |
| $H_2S$ | -0.4238 | 0.2360 | -1.7958 | 0.078 | | $PON:NO_3^-$ | -0.3501 | 0.0624 | -5.6093 | <0.001 |
| $PON:NO_3^-$ | -0.3568 | 0.0593 | -6.0169 | <0.001 | | $PON:NO_2^-$ | 4.3858 | 0.6737 | 6.5103 | <0.001 |
| $PON:NO_2^-$ | 3.7572 | 0.6650 | 5.6499 | <0.001 | | $NO_3^-:NO_2^-$ | -3.5869 | 0.8114 | -4.4207 | <0.001 |
| $NO_3^-:NO_2^-$ | -2.4176 | 0.6895 | -3.5065 | <0.001 | | $NO_3^-:O_2$ | -0.0434 | 0.0122 | -3.5715 | <0.001 |
| $NO_3^-:O_2$ | -0.0350 | 0.0107 | -3.2777 | 0.002 | | $NO_3^-:O_2$ | 0.1829 | 0.0570 | 3.2073 | 0.002 |
| $NO_3^-:H_2S$ | 0.3773 | 0.1476 | 2.5571 | 0.013 | | | | | | |
| $NO_2^-:H_2S$ | -2.5429 | 0.8945 | -2.8429 | 0.006 | | | | | | |
| $O_2:H_2S$ | 0.0183 | 0.0090 | 2.0350 | 0.047 | | | | | | |
| | | | $R^2$ | **0.8109** | | | | | $R^2$ | **0.7666** |

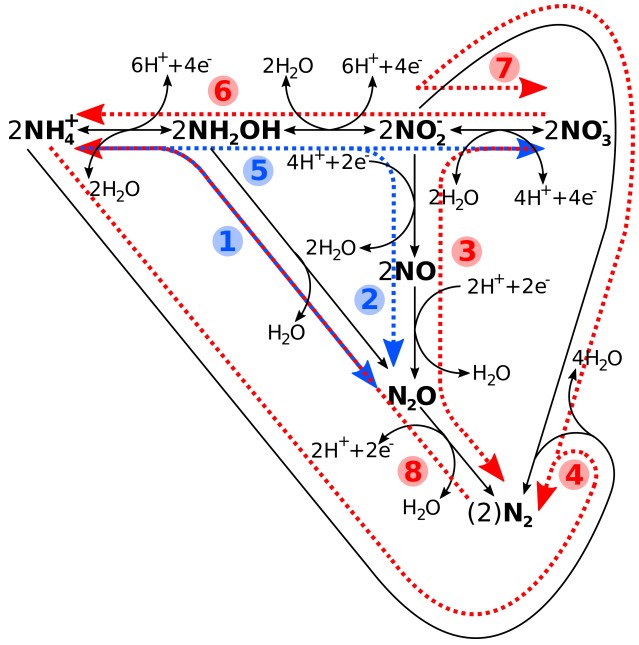

**Figure 1.** Schematic reaction diagram of major marine nitrogen cycling processes, with (1) Hydroxylamine Oxidation, (2) Nitrifier Denitrification, (3) Denitrification, (4) Anammox, (5) Nitrification, (6) Nitrate Ammonification, aka DNRA, Anaerobic Nitrite Oxidation (7) and (8) Nitrogen Fixation. While processes (1–4) are considered nitrogen loss processes, (5–7) constitute neither a loss nor a gain, with (8) being the latter. Blue colours denote oxic and red suboxic/anoxic processes. Please note that the reactions have been chemically balanced and for electroneutrality, the exact amount of $H^+$, $e^-$ or water produced/consumed will depend on the actual organism/enzyme and reaction pathway. See Bourbonnais et al. (2017); Codispoti et al. (2005); Zumft (1997) and references therein for details.

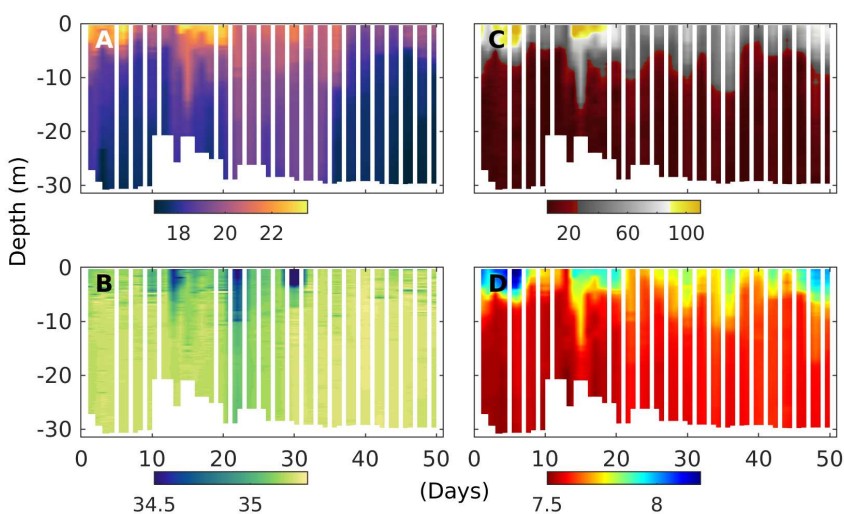

**Figure 2.** Temporal development of upwelling and stratification in the surrounding Pacific waters at the mooring site (water depth between 25 and 30 metres) as evidenced by changes in CTD-derived **(A)** temperature (°C), **(B)** salinity (psu), **(C)** oxygen saturation (%) and **(D)** pH$_T$ (total scale) depth profiles. Note that dips in salinity at the surface correspond with El Niño related torrential rain events further land inwards and increased discharge of freshwater by the nearby river Rimac.

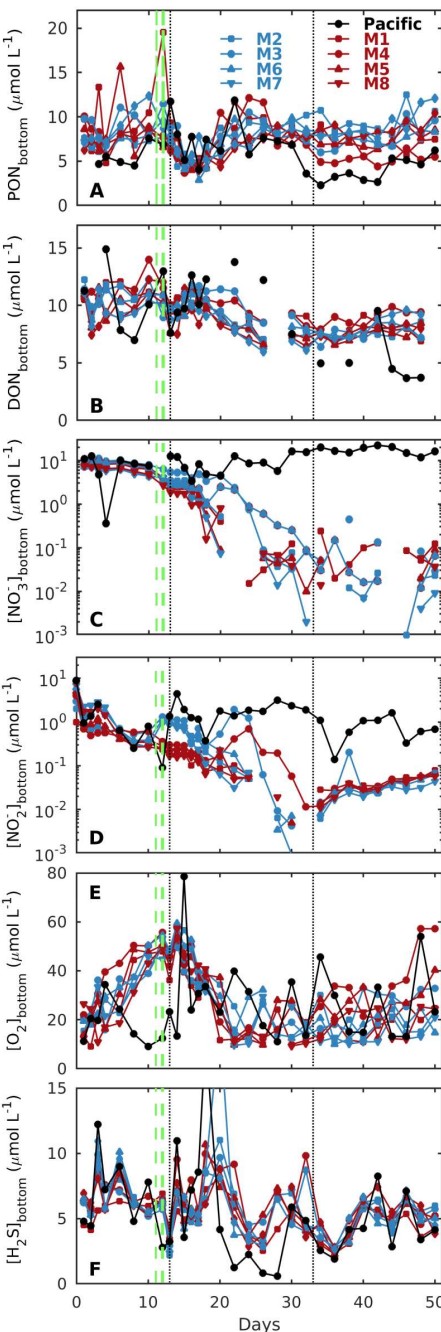

**Figure 3.** Temporal evolution of depth-integrated bottom layer (**A**) PON, (**B**) DON, (**C**) nitrate ($NO_3^-$), (**D**) nitrite ($NO_2^-$), together with CTD-derived (**E**) oxygen ($O_2$) and (**F**) hydrogen sulphide ($H_2S$) concentrations at the Niskin sampling depth, in the mesocosms (M1-M8) and the surrounding Pacific. Blue and red colours denote the 'low N/P' and 'very low N/P' deep water additions, respectively (see section 2.2 for details). The dashed green lines denote deep water additions on days 11 and 12, while the dotted black lines denote additions of a brine solution to the bottom layer to increase salinity, strengthen stratification and reduce mixing. See Methods for details.

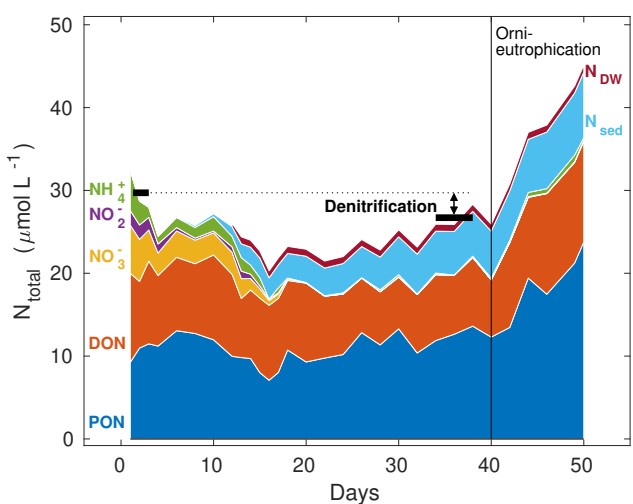

**Figure 4.** Representative example of a total nitrogen budget from mesocosm M7, considering all relevant pools such as particulate organic nitrogen (PON), dissolved organic nitrogen (DON), dissolved inorganic nitrogen in the form of nitrate ($NO_3^-$), nitrite ($NO_2^-$) and ammonium ($NH_4^+$), cumulative particulate organic nitrogen exported to the sediment trap ($N_{sed}$) and the net change to all the above mentioned nitrogen species (with the exception of PON for which there was no deep water data) by deep water addition ($N_{DW}$). Black horizontal markers denote the total amount of nitrogen in these pools, calculated as an average of three consecutive sampling days at the start of the experiment (days 1-3) and towards the end (days 36-40), prior to the onset of orni-eutrophication. The deficit in this N-budget comprises all nitrogen loss processes, dominated by denitrification (compare Tab. 2). Please note that the initial dip in nitrogen inventory is probably the result of a lag phase of nitrogen settling into the sediment trap.

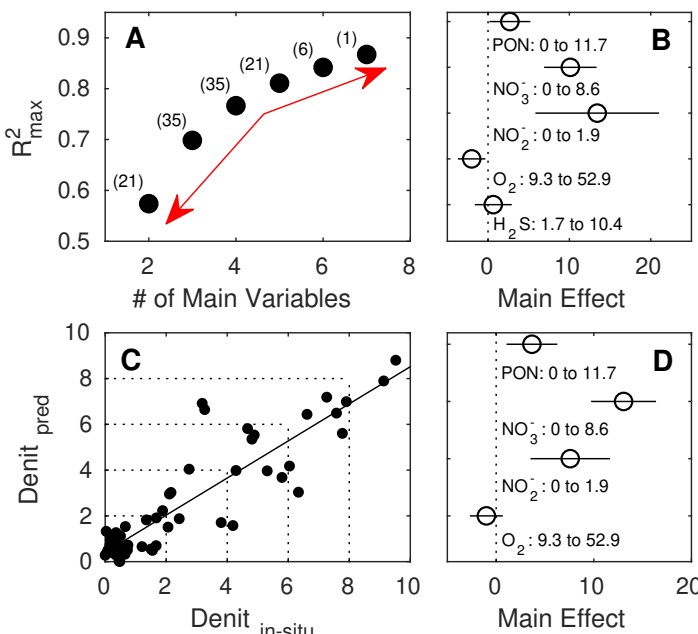

**Figure 5.** Stepwise multiple linear regressions (MLRs) of in-situ denitrification rates in the mesocosms against up to 7 potential predictors and their interactions (PON, $PON_{sed}$, DON, $NO_3^-$, $NO_2^-$, $O_2$ and $H_2S$), with (**A**) reducing numbers of measured variables and resulting maximum $R^2$. The red arrows denote the relatively small increase in maximum $R^2$ beyond 5 main variables and the relatively large decrease below, indicating this to be the sweet-spot in terms of balancing model complexity with predictive power, avoiding overfitting (see Methods for details). Numbers in brackets denote the number of possible predictor combinations, i.e. MLRs fitted. (**B**) Main effect sizes of the stepwise MLR with five main variables and highest (0.8109) $R^2$ (compare Tab. 3). (**C**) Linear fit through in-situ and predicted denitrification rates (nmol $N_2$ $L^{-1}h^{-1}$) by the MLR shown in (**D**). For comparison, main effect sizes of the stepwise MLR with four main variables and highest (0.7666) $R^2$ (compare Tab. 3).

**Appendix A**

**Table A1.** Measured denitrification rates ($nmol\,N_2\,L^{-1}h^{-1}$) in the low N/P and very low N/P deep water addition incubations and the surrounding Pacific, at various days. The average uncertainty, calculated from the standard error of the regression slope, was 20% of measured rates. Numbers in brackets refer to rates that exceeded the theoretical maximum, based on substrate availability.

| Mesocosm | Measured denitrification rates | | | | | | | | | |
|---|---|---|---|---|---|---|---|---|---|---|
| | | | | | ($nmol\,N_2\,L^{-1}h^{-1}$) | | | | | |
| low N/P | T8 | T12 | T16 | T22 | T26 | T30 | T34 | T38 | T42 | T46 |
| M2 | 14.99 | 7.77 | 1.96 | (28.92) | (18.51) | (37.20) | (15.76) | (15.4) | (19.93) | (8.05) |
| M3 | 4.70 | 10.74 | 10.23 | 8.51 | (27.46) | (28.93) | (22.08) | 21.24 | (20.56) | (7.87) |
| M6 | 3.86 | 20.46 | 3.32 | 7.08 | (16.74) | (16.93) | (14.37) | (16.24) | (12.08) | 3.02 |
| M7 | 5.94 | 3.92 | 4.62 | (14.15) | (13.81) | (20.91) | (22.64) | (13.08) | (10.90) | (10.18) |

| very low N/P | | | | | | | | | | |
|---|---|---|---|---|---|---|---|---|---|---|
| M1 | 6.68 | 18.65 | 13.8 | (32.34) | (30.62) | (24.79) | (29.32) | 0.89 | 3.9 | 7.39 |
| M4 | 0.74 | 43.33 | 11.46 | 7.51 | 23.78 | (28.79) | (4.64) | 2.47 | 1.35 | 0.32 |
| M5 | 5.33 | 82.23 | 4.05 | (25.34) | (17.30) | (5.55) | (13.41) | (12.68) | (16.21) | 3.36 |
| M8 | 10.81 | 7.85 | 4.67 | (13.23) | (6.67) | (22.83) | (17.24) | (7.21) | (9.78) | 1.00 |

| PACIFIC | 9.19 | (46.52) | 7.45 | 18.12 | 0.10 | 2.72 | 0.22 | 0.00 | 0.26 | 50.52 |

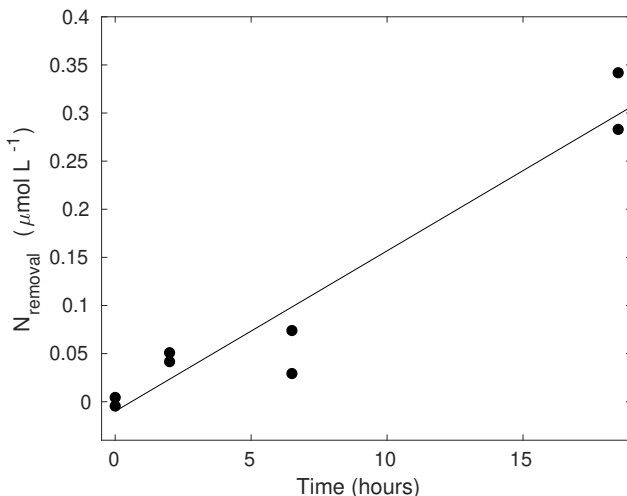

**Figure A1.** Example of nitrogen removal during label incubations (Pacific on Day 8) as calculated according to Eq. 1. Denitrification or anammox rates were calculated from the slope of a linear fit to the data. It is acknowledged that in some samples a lag-phase was encountered.

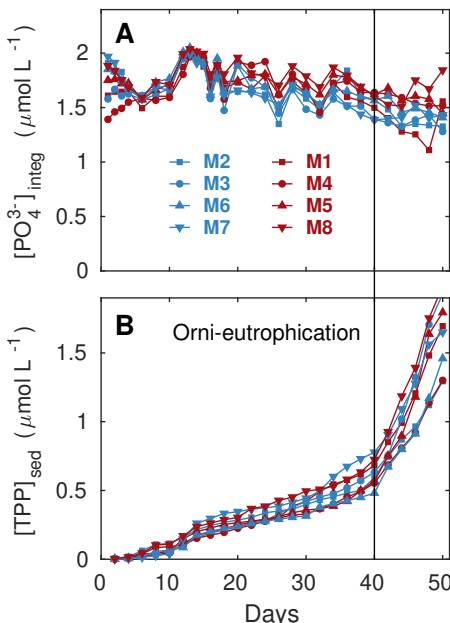

**Figure A2.** Temporal evolution of depth-integrated (0-17 m) phosphate concentrations in the mesocosms (**A**), together with total particulate phosphorus (TPP) accumulating in the sediment traps (**B**). The black vertical line marks the onset of orni-eutrophication. Blue and red colours denote the 'low N/P' and 'very low N/P' deep water additions, respectively (see section 2.2 for details) in the various mesocosms (M1-M8).

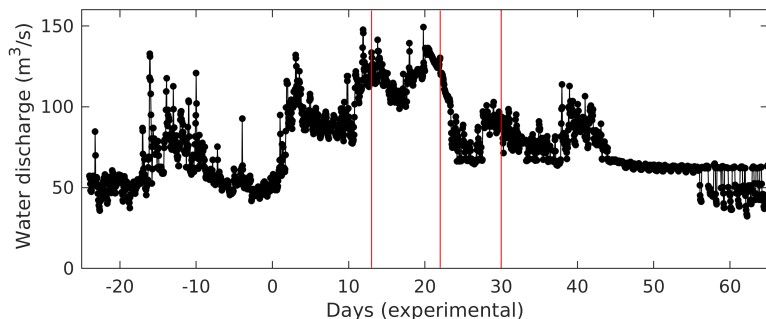

**Figure A3.** Water discharge rates at station Chosica in the Rimac, ∼45 km upstream the river mouth, and the latter being ∼5 nm from the mesocosm mooring site. Red lines denote experimental days at which significant reductions in surface water salinities around the mesocosms were measured (compare Fig. 2). Note that the third measured surface freshening does not seem to coincide with such high discharge rates as for the first two. The freshwater could therefore stem from the River Chillón, ∼9 nm to the North.

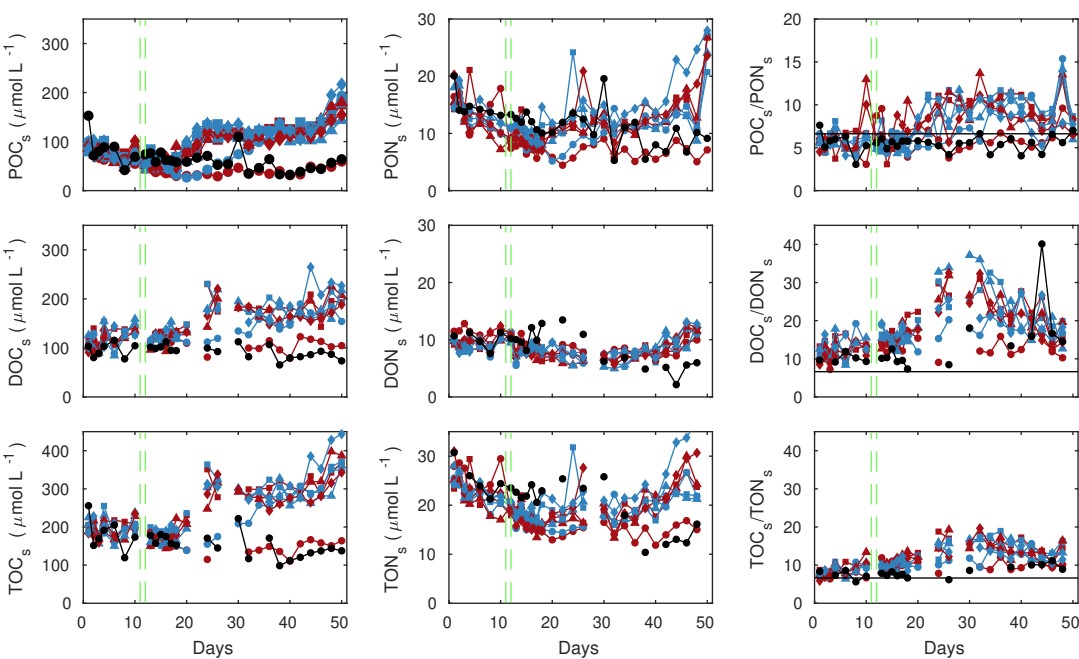

**Figure A4.** Particulate, dissolved and total organic carbon and nitrogen in the mesocosms' surface waters, and resulting ratios. Style and colour code follow those shown in Fig. A2, while black shows data for the surrounding Pacific

.