# Peer review of "Nitrogen loss processes in response to upwelling in a Peruvian coastal setting dominated by denitrification – a mesocosm approach"

_Biogeosciences, 2021_

## Referee Comment (RC1)

**Review of Schulz *et al.***

**General comments**
Schulz et al. present a mesocosm study in the Eastern Upwelling System off Peru. The mesocosm approach is relatively novel for studies of this type, although it suffers from its own limitations (e.g., orni-eutrophication). The manuscript is concisely written and proposes that roughly half of the nitrogen that could be exported to depth is denitrified in coastal waters during coastal upwelling. While this study is worthy of publication, I do have some concerns, mostly regarding purging of the samples during the $^{15}N$-labeled incubations, which removes $H_2S$ - an electron donor during chemolithoautotrophic denitrification. Furthermore, since mostly all $O_2$ is removed during purging with He prior to incubating the samples, their measured rates are not representative of *in-situ* conditions ($O_2$ concentrations in bottom waters were generally higher than 20 µmol $L^{-1}$, which is too high for $N_2O$ conversion to $N_2$ (see Dalsgaard et al., 2014)).

**Specific comments**

**Line 9:** "This was in contrast to realized rates in the surrounding Pacific". This sentence is unclear as the author later claim that rates measured as part of the mesocosm experiment and in surrounding waters were comparable.

**Lines 39-43:** This sentence about ocean acidification seems a bit out of place since the connection with OMZs and nitrogen cycling is not clear. I suggest removing.

**Line 73:** How long were the samples stored before analysis?

**Line 74:** $O_2$ concentrations should ideally be monitored during $^{15}N$-labeled incubations using non-invasive $O_2$ measurement technology, such as Oxysense (http://www.oxysense.com/), Pyroscience (http://www.pyroscience.com/) or custom-made sensors (e.g., see Larsen *et al*., 2016). This is also to ensure that no $O_2$ is infiltrating during the incubations from the stoppers.

**Line 75:** I think such a high $O_2$ offset is problematic and not representative of anoxic coastal waters off Peru – where $O_2$ concentrations are generally well below 10 µM. I suspect that $O_2$ is also introduced during sampling from a Niskin bottle (de Brabandere et al.. 2012). Furthermore, $O_2$ in the nanomolar range has been shown to influence $N_2$ production rates (Dalsgaard et al., 2014). Since their samples are purged with He, the estimated rates are only putative and likely not representative of real *in-situ* conditions. I understand the limitation, but *in-situ* incubations would be preferable.

**Lines 80-82:** Why is this treatment referred to as a moderate treatment if the N:P values between this and the extreme treatment are similar? I would rename this treatment as it is clearly not representative of moderate N loss conditions.

**Lines 82-83:** How did they prevent gas exchange and minimize $O_2$ contamination in these waters during collection/injection? As this is an important detail for their experiment, a brief explanation should be added here (without having to refer to the Bach et al. (2020) paper).

**Lines 87-88:** Why not use $^{15}$N-labeled $NO_3^-$ to measure denitrification rates? Nitrate concentrations are generally higher than $NO_2^-$ and $NH_4^+$ (thus, the substrate is less limiting).

**Line 89:** Why 3 µmol L-1? It seems to be a bit arbitrary.

**Lines 86-93:** On a cautionary note, other studies (de Brabandere et al., 2013; Chang et al., 2014), observed that more $^{29}N_2$ is sometimes produced than could be accounted for assuming a binomial distribution after taking the production due to anammox. They propose that "nitrite shunting" where $NO_3^-$ is converted to $N_2$ completely intracellularly without exchange with the external ambient $NO_2^-$ pool could lead to that $^{29}N_2$ excess. I am curious to know if such $^{29}N_2$ excess was also observed in this study. Using $NO_3^-$ as a tracer could help to account for this process.

**Lines 93-101:** I understand that purging with He is necessary since these are anoxic incubations, but since $H_2S$ was present in bottom waters, removing all gases (including $H_2S$) would underestimate chemoautotrophic denitrification. I strongly recommend complementation of stripped gases involved in N cycling metabolisms.

**Line 99:** What is the recommended flow rate? I assume purging time is adjusted to leave enough background $N_2$ for GC-IRMS measurements?

**Lines 102-103:** At what temperatures were samples equilibrated?

**Lines 110-114:** This approach could potentially affect their rate calculations if "nitrite shunting" produces $^{29}N_2$ excess. See above comment (lines 86-93).

**Lines 113-115:** In figure A1, it seems like there might be an exponential increase from time point 6 hrs and 20 hrs. In this case, I would only use the linear portion for rate calculations. Is this observed for all other incubations? It would have been useful to obtain another time point somewhere in between (at 12 hrs) to better disregard this possibility.

**Lines 134-138:** A more correct approach would be to construct Michaelis-Menten curves and calculate the half saturation constants and maximum denitrification rates from the measured *in-situ* rates (see Michiels et al., 2019).

**Lines 146-147:** Change to: "24 hrs per day x 38 days x 2 (conversion between $N_2$ to N) divided by 3 (contribution of bottom layer water to overall mesocosm volume), ..."

**Lines 189-191:** The oxygen concentrations shown in Fig. 3E are generally above 20 µmol L$^{-1}$, which would be too high for $N_2O$ conversion to $N_2$ (see Dalsgaard et al., 2014 and Frey et al., 2020). Therefore, their measured rates are potential and denitrification was likely only observed because the samples were purged with $O_2$ - removing mostly all $O_2$.

**Line 194:** I don't think it make sense to call this a "moderate" treatment (see above comment lines 80-82).

**Lines 199-200:** Again, these relatively high $H_2S$ concentrations indicate that chemoautotrophic denitrification might be an important process that was not measured (since samples were purged with He before the incubations).

**Lines 211-214:** The near perfect agreement between the two approaches is a bit surprising considering that measured rates are potential and likely not representative of *in-situ* conditions.

**Lines 221-223:** Were $N_2$ fixation rates measured at the same depths than for the denitrification/anammox incubations in the mesocosms?

**Line 266:** The calculated overall nitrogen loss could also be overestimated since *in-situ* denitrification rates were likely lower. Samples collected in the mesocosms and surrounding Pacific waters were purged before the incubations, removing mostly $O_2$ and thus creating conditions more conducive to $N_2$ loss. The $O_2$ concentrations observed in bottom waters were too high for $N_2O$ conversion to $N_2$ (see Dalsgaard et al., 2014).

**Lines 313:** Why did C/N values not increase in that one mesocosm?

**Lines 319-321:** It is also possible that the measured DON pool was mostly recalcitrant, with fast cycling of labile DON.

**Lines 340-341:** Denitrification/anammox linked to microenvironments around particles would not be captured by $^{15}N$-labeled incubations, especially if these are not performed *in situ*.

**Lines 343-345:** It is unclear how $H_2S$ would inhibit anammox in their incubations, since samples were purged (hence $H_2S$ was removed).

**Lines 350-353:** It would be relevant to include these data (i.e., anammox functional marker gene *hzo*) in the manuscript.

**Line 375:** Why is the contribution from the Arabian Sea, where significant N-loss occurs, is not taken into account here?

**Lines 376-379:** I don't think there is anything new in this statement. Due to the large uncertainties associated with these estimates, it is still unclear if the majority of the N-loss occurs in the water-column or sediments.

**Lines 379-380:** Why is export production projected to decrease if upwelling intensity and frequency (and thus nutrient supply) is expected to increase (Hauri et al., 2013 and Wang et al., 2015 papers cited in the introduction)?

**Table 2:** I would rename the "moderate" treatment to "extreme" since the degree of N-loss is similar in both treatments.

It is odd to express individual N-budgets for each mesocosms as negative values and present the mean as a positive value. I suggest renaming these columns N-loss from $^{15}$N-labeled incubations and N-loss from N-budget.

**Figure 2:** What was bottom depth at the mooring site?

**Figure 1A:** It is difficult to tell if the last time point around t = 20 hrs represents an exponential increase (as often observed for $^{15}$N-labeled incubations).

**Technical corrections**

**Line 49:** define nm (i.e., nautical miles)

**Line 63:** change acoording to "according"

**Additional references:**

Dalsgaard, T., Stewart, F. J., Thamdrup, B., De Brabandere, L., Revsbech, N. P., Ulloa, O., ... & DeLong, E. F. (2014). Oxygen at nanomolar levels reversibly suppresses process rates and gene expression in anammox and denitrification in the oxygen minimum zone off northern Chile. *MBio*, *5*(6).

De Brabandere, L., Thamdrup, B., Revsbech, N. P., & Foadi, R. (2012). A critical assessment of the occurrence and extend of oxygen contamination during anaerobic incubations utilizing commercially available vials. *Journal of microbiological methods*, *88*(1), 147-154.

Larsen, M., Lehner, P., Borisov, S. M., Klimant, I., Fischer, J. P., Stewart, F. J., Canfield, D. E., & Glud, R. N. (2016). In situ quantification of ultra-low O2 concentrations in oxygen minimum zones: Application of novel optodes. *Limnology and Oceanography: Methods*, *14*(12), 784-800.

---

## Author Comment (AC1)

**Response to Reviewer #1**

We thank Reviewer #1 for their insightful comments and suggestions. Below please find our detailed point-by-point response.

**General Comments**

**1.** '... While this study is worthy of publication, I do have some concerns, mostly regarding purging of the samples during the 15 N-labelled incubations, which removes $H_2S$ - an electron donor during chemolithoautotrophic denitrification. Furthermore, since mostly all $O_2$ is removed during purging with He prior to incubating the samples, their measured rates are not representative of in-situ conditions ($O_2$ concentrations in bottom waters were generally higher than 20 $\mu molL^{-1}$, which is too high for $N_2O$ conversion to $N_2$....'

    RESPONSE: We will address the issues raised by the reviewer in the specific responses below.

**Specific comments of Reviewer #1**

**Line 9:** This was in contrast to realized rates in the surrounding Pacific". This sentence is unclear as the author later claim that rates measured as part of the mesocosm experiment and in surrounding waters were comparable.

    RESPONSE: We will clarify this statement with 'In the surrounding Pacific measured denitrification rates were similar, although no indications of substrate limitation were detected.'

**Lines 39-43:** This sentence about ocean acidification seems a bit out of place since the connection with OMZs and nitrogen cycling is not clear. I suggest removing.

    RESPONSE: We will clarified the connection by adding the following 'Changes in upwelling frequency/intensity, oxygen availability, temperature and pH can influence planktonic food web functioning in EBUS, with repercussions for nitrogen loss processes.'

**Line 73:** How long were the samples stored before analysis?

    RESPONSE: It was about 2-4 hours. This information will be added.

**Line 74:** $O_2$ concentrations should ideally be monitored during 15N-labelled incubations using non-invasive $O_2$ measurement technology .... This is also to ensure that no $O_2$ is infiltrating during the incubations from the stoppers.

    RESPONSE: We did not monitor the oxygen inside the exetainers in this experiment. However, data from other very similar exetainer experiments, in which $O_2$ was monitored with the Lumos sensors (Sun et al. 2020), show that the $O_2$ level varied between essentially zero and up to 100 nM over a period of two days. Even that highest concentration is below the lowest thresholds reported to inhibit anammox and denitrification (>200 nM Dalsgaard et al. 2014; see discussion in the manuscript, section 4.2.2).

**Line 75:** I think such a high $O_2$ offset is problematic and not representative of anoxic coastal

waters off Peru – where $O_2$ concentrations are generally well below 10 $\mu$M. I suspect that $O_2$ is also introduced during sampling from a Niskin bottle (de Brabandere et al.. 2012). Furthermore, $O_2$ in the nanomolar range has been shown to influence $N_2$ production rates (Dalsgaard et al., 2014). Since their samples are purged with He, the estimated rates are only putative and likely not representative of real in-situ conditions. I understand the limitation, but in-situ incubations would be preferable.

RESPONSE: The reviewer is correct, it would be wonderful to be able to do in situ incubations in order to assess true rates of denitrification and anammox. Unfortunately, at present, we are not aware of any realistic way to do that. The in situ incubation methods are cumbersome and allow at most a few measurements per day (e.g., Collins et al. 2018, Ward et al. 2019), and at present would be difficult to deploy in a mesocosm and hard to scale up to the numbers of experiments required for the experimental design used here. The non incubation approach of measuring in situ gas concentrations (gas tension device, Reed et al. 2018) has great attractions. This method was published after the study under review had been completed. The GTD has not been widely used yet, but based on the presentation by Altabet et al. at the Ocean Sciences meeting in 2020, it promises to be very useful on the oceanographic scale, although with substantial assumptions and constraints. So much as we would like to, making in situ rate measurements in the mesocosms is not yet feasible.

The purging approach reduces the gas concentrations for all gases prior to the incubations (except helium), including $N_2, N_2O, O_2, CO_2, H_2S$, etc. By lowering the oxygen concentration, purging may enhance the rates of anaerobic processes such as denitrification and anammox. But as observed by de Brabandere et al. (2013), the fact that the processes occurred in anoxic incubations in water collected from oxic layers indicates the presence of viable populations of microbes capable of those processes. That implies the definite potential for in situ activities.

We have also re-calculated oxygen concentrations as measured by the optical CTD sensor by applying a 1 second response time hysteresis correction as described in Fiedler et al. (2013), rather than just mentioning the issue that 'raw' CTD data will over-estimate oxygen concentrations in a particular depth during a down cast, when moving from high to low oxygen concentrations. Now, corrected oxygen concentrations at the sampling depth are hovering around 20 $\mu$molL$^{-1}$ for most of the time. Together with the +13 $\mu$molL$^{-1}$ offset in comparison to Winkler titrations in these samples, this suggests that in-situ oxygen concentrations reached indeed below 10 $\mu$molL$^{-1}$, being representative for anoxic coastal waters off Peru.

**Lines 80-82:** Why is this treatment referred to as a moderate treatment if the N:P values between this and the extreme treatment are similar? I would rename this treatment as it is clearly not representative of moderate N loss conditions.

RESPONSE: We will re-name the treatments to 'low N/P' and 'very low N/P', respectively, which will also keep it consistent with the terminology introduced in the accompanying paper by Bach et al. (2020).

**Lines 82-83:** How did they prevent gas exchange and minimize $O_2$ contamination in these

waters during collection/injection? As this is an important detail for their experiment, a brief explanation should be added here (without having to refer to the Bach et al. (2020) paper).

RESPONSE: We will add the following piece of information: 'The deep waters were collected into 100 $m^3$ bags at the respective depths and sealed once brought back to the surface. Deep water was added by first removing about 20 $m^3$ from each mesocosm and replacing it with the respective deep water that was injected into the bottom layer between 14-17 metres on day 11, and the surface layer between 1-9 metres on day 12. To minimise changes to deep water gas concentrations during injections, water was pumped from several meters depth out of the deep water bags.'

**Lines 87-88:** Why not use 15 N-labeled $NO_3^-$ to measure denitrification rates? Nitrate concentrations are generally higher than $NO_2^-$ and $NH_4^+$ (thus, the substrate is less limiting).

RESPONSE: Previous experience has shown repeatedly that lower rates of $N_2$ production result from parallel incubations with $^{15}NO_3^-$ vs $^{15}NO_2^-$. We have interpreted that difference to the exchange of $NO_2^-$ as an intermediate in the complete denitrification pathway, i.e. $^{15}NO_2^-$ produced from $^{15}NO_3^-$ is diluted with residual $^{14}NO_2^-$ in the medium and reduces the amount of label that makes it all the way to $N_2$. Thus, the rates measured from $^{15}NO_2^-$ are a better estimate of the actual rate of $N_2$ production. The fact that we routinely detect $^{15}NO_3^-$ reduction (as $^{15}NO_2^-$ production) in these same incubations shows that $NO_2^-$ dilution does indeed occur.

**Line 89:** Why 3 $\mu mol L^{-1}$? It seems to be a bit arbitrary.

RESPONSE: 3 $\mu mol L^{-1}$ tracer addition is pretty standard. For $NO_2^-$, in particular in ODZ conditions, it is a level commonly occurring in natural waters, so does not represent a big perturbation, but ensures enough substrate to obtain a signal in the product. At the beginning of the experiment similar concentrations were present in the bottom layer. However, being closed systems, $NO_2^-$ concentrations decreased over time, so the tracer addition could have stimulated measured rates, in particular after the N-depleted deep water addition. This is consistent with the statements in L134-138 (original MS) about the measured rates exceeding the rates that would be possible at in situ substrate concentration.

**Lines 86-93:** On a cautionary note, other studies (de Brabandere et al., 2013; Chang et al., 2014), observed that more $^{29}N_2$ is sometimes produced than could be accounted for assuming a binomial distribution after taking the production due to anammox. They propose that "nitrite shunting" where $NO_3^-$ is converted to $N_2$ completely intracellularly without exchange with the external ambient $NO_2^-$ pool could lead to that $^{29}N_2$ excess. I am curious to know if such $^{29}N_2$ excess was also observed in this study. Using $NO_3^-$ as a tracer could help to account for this process.

RESPONSE: As we have only made incubations with labelled $NO_2^-$ we have no comparison, or binomial expectation. Please also refer to our response to the Lines 87-88 comment.

**Lines 93-101:** I understand that purging with He is necessary since these are anoxic incubations, but since $H_2S$ was present in bottom waters, removing all gases (including $H_2S$) would underestimate chemoautotrophic denitrification. I strongly recommend complementation of

stripped gases involved in N cycling metabolisms.

RESPONSE: The reasoning behind He purging is to reduce background $^{14}N_2$ concentrations (to more easily pick up label incorporation into the $N_2$ produced during denitrification or anammox), not to strip the solution off dissolved gases. And indeed, we saw in earlier studies that we can observe massive rates of chemoautotrophic denitrification using this approach (Kalvelage et al., 2013), suggesting that a significant portion of dissolved gases such as $H_2S$ is retained. Hence, this method doesn't appear to necessarily underestimate rates resulting from He bubbling.

**Line 99:** What is the recommended flow rate? I assume purging time is adjusted to leave enough background $N_2$ for GC-IRMS measurements?

RESPONSE: As mentioned previously, the idea of purging is to reduce background $N_2$ in order to enhance the ability to detect the small isotopic signal of the labelled $N_2$ produced during the incubation. Since $N_2$ concentration doesn't vary much in seawater at the level we can detect with these methods, we used a standard flow rate (monitored by assuring $1-2$ psi at the exetainer level, rather than flow rate at the cylinder level) and purging time, previously calculated to assure at least 10-fold replacement volumes for the exetainers. These methods have been described in great detail elsewhere by ourselves and others, which is why we did not repeat all the details and rationale here.

**Lines 102-103:** At what temperatures were samples equilibrated?

RESPONSE: Samples were equilibrated at room temperature. This will be mentioned in the text.

**Lines 110-114:** This approach could potentially affect their rate calculations if "nitrite shunting" produces $^{29}N_2$ excess. See above comment (lines 86-93).

RESPONSE: Please see our response to the previous comment(s).

**Lines 113-115:** In figure A1, it seems like there might be an exponential increase from time point 6 hrs and 20 hrs. In this case, I would only use the linear portion for rate calculations. Is this observed for all other incubations? It would have been useful to obtain another time points somewhere in between (at 12 hrs) to better disregard this possibility.

RESPONSE: Given the clear increase between 0 and 2 hours, we would rather argue for the lower 7 hour data point to be off. Furthermore, deviations from a linear increase over the entire incubation period was rather observed at times to be resulting from a lag phase during the first 2 hours, as acknowledged in the figure caption. We agree that having an extra time point between 8 and 20 hours would have been ideal in hindsight, but there are always logistical constraints. Furthermore, having the majority of samples in the first half of the incubation proved to be quite valuable as the lag phase issue could be clearly detected and accounted for, when necessary. Finally, restricting the incubation period to just under one day ensured to avoid what is usually considered to introduce potential bottle-effects. We will make reference to the latter issue in the methods section.

**Lines 134-138:** A more correct approach would be to construct Michaelis-Menten curves and calculate the half saturation constants and maximum denitrification rates from the measured in-situ rates (see Michiels et al., 2019).

RESPONSE: There is surprisingly little information on the dependence of denitrification on the concentration of nitrate or nitrite. This is probably because denitrification is often shown to be limited by organic matter concentration in oceanic systems. Again, in the coastal mesocosms, this may not have been the case – OM may not have been the limiting factor – and as noted above, the tracer addition could have stimulated measured rates. Lacking direct experiments on the MM kinetics of denitrification in this system, the approach used here is probably a reasonable compromise.

**Lines 146-147:** Change to: "24 hrs per day x 38 days x 2 (conversion between N 2 to N) divided by 3 (contribution of bottom layer water to overall mesocosm volume), ..."

RESPONSE: We agree and will make the suggested changes.

**Lines 189-191:** The oxygen concentrations shown in Fig. 3E are generally above 20 $\mu$molL$^{-1}$, which would be too high for $N_2O$ conversion to $N_2$ (see Dalsgaard et al., 2014 and Frey et al., 2020). Therefore, their measured rates are potential and denitrification was likely only observed because the samples were purged with helium - removing mostly all $O_2$.

RESPONSE: We agree, like in any assay incubation, measured rates should be taken with a grain of salt when extrapolating to in-situ conditions. Please also see our replies on oxygen response time hysteresis correction (suggesting oxygen levels below 10 $\mu$molL$^{-1}$ for most of the experiment), as well as helium purging.

**Line 194:** I don't think it makes sense to call this a "moderate" treatment (see above comment lines 80-82).

RESPONSE: As suggested, we will change the terminology.

**Lines 199-200:** Again, these relatively high $H_2S$ concentrations indicate that chemoautotrophic denitrification might be an important process that was not measured (since samples were purged with He before the incubations).

RESPONSE: Please see our response to the Lines 93-101 comment. Furthermore, please also see our response to the L226 comment from reviewer #2 on nitrite and organic matter being the main drivers of denitrification.

**Lines 211-214:** The near perfect agreement between the two approaches is a bit surprising considering that measured rates are potential and likely not representative of in-situ conditions.

RESPONSE: Good point. We have wondered about that ourselves. Considering the many caveats for incubation-based rate measurements, how can they make so much sense in comparison to processes estimated from several other independent in-situ measurements? The only conclusion we can come up with is that the rates measured in the essay incubations must have been similar to what was happening in-situ.

**Line 266:** The calculated overall nitrogen loss could also be overestimated since in-situ denitrification rates were likely lower. Samples collected in the mesocosms and surrounding Pacific waters were purged before the incubations, removing mostly $O_2$ and thus creating conditions more conducive to $N_2$ loss. The $O_2$ concentrations observed in bottom waters were too high for $N_2O$ conversion to $N_2$ (see Dalsgaard et al., 2014).

RESPONSE: Please see our responses to a number of comments above, in particular the one on the new CTD-oxygen-optode response time correction - Line 75 comment.

**Lines 313:** Why did C/N values not increase in that one mesocosm?

RESPONSE: Deep water additions were followed by a bloom of the dinoflagellate *Akashiwo sanguinea*, fixing carbon without significant net nitrogen assimilation, in all except this one mesocosm. We will mention this fact, which is described in more detail in the accompanying Bach et al. (2020) paper, in the discussion.

**Lines 319-321:** It is also possible that the measured DON pool was mostly recalcitrant, with fast cycling of labile DON.

RESPONSE: We agree and, as stated in the text, it would require preferential N over C remineralisation.

**Lines 340-341:** Denitrification/anammox linked to microenvironments around particles would not be captured by 15N-labeled incubations, especially if these are not performed in situ.

RESPONSE: As the incubation seawater was not filtered and hence contained particles, microenvironments around those are likely to have been re-established during the 20 hours of incubation.

**Lines 343-345:** It is unclear how $H_2S$ would inhibit anammox in their incubations, since samples were purged (hence $H_2S$ was removed).

RESPONSE: Please see our responses to various comments above.

**Lines 350-353:** It would be relevant to include these data (i.e., anammox functional marker gene hzo) in the manuscript.

RESPONSE: Unfortunately, no genomic data can be presented at this stage, as of ongoing Nagoya Protocol negotiations. However, as there is no discrepancy between our rate measurements and described gene abundance observation, there shouldn't be a need to explicitly show the latter.

**Line 375:** Why is the contribution from the Arabian Sea, where significant N-loss occurs, not taken into account here?

RESPONSE: We will include reference to the Arabian Sea and Bay of Bengal in a revised version of our manuscript. This will not change the main findings and conclusions of our calculations.

**Lines 376-379:** I don't think there is anything new in this statement. Due to the large uncertainties associated with these estimates, it is still unclear if the majority of the N-loss occurs in the water-column or sediments.

RESPONSE: The discussion in the last paragraph is only about water column denitrification, i.e. a comparison of globally assembled in-situ estimates with our mesocosm-derived measurements.

**Lines 379-380:** Why is export production projected to decrease if upwelling intensity and frequency (and thus nutrient supply) is expected to increase (Hauri et al., 2013 and Wang et al., 2015 papers cited in the introduction)?

RESPONSE: This is a valid point raised by the reviewer. We will be more specific here and explain that projected reductions in global export production are thought to result from changes to community structure. Furthermore, in regards to export production in ODZs and OMZs, as of a potentially counter-acting increase in upwelling intensity and frequency, we will be more cautious about the expected sign of change.

**Table 2:** I would rename the "moderate" treatment to "extreme" since the degree of N-loss is similar in both treatments. It is odd to express individual N-budgets for each mesocosms as negative values and present the mean as a positive value. I suggest renaming these columns N-loss from 15N-labelled incubations and N-loss from N-budget.

RESPONSE: We agree and will change the terminology to 'low' and 'very low' N/P. We will also change the N-budget mean to a negative number to match the individual mesocosm values.

**Figure 2:** What was bottom depth at the mooring site?

RESPONSE: The depth at the mooring site was between 25 to 30 metres. This information will be added to the figure caption.

**Figure 1A:** It is difficult to tell if the last time point around t = 20 hrs represents an exponential increase (as often observed for 15N-labelled incubations).

RESPONSE: Please see our response to the Lines 113-115 comment.

**Technical corrections**

**Line 49:** define nm (i.e., nautical miles).

RESPONSE: We agree.

**Line 63:** change acoording to "according"

RESPONSE: We will do.

---

## Author Comment (AC2)

**Response to Reviewer #2**

We thank Reviewer #2 for their insightful comments and suggestions. Below please find our detailed point-by-point response.

**General Comments**

**1.** It would help if the authors identified more clearly the goal of the study. They say it was to "quantify the importance of nitrogen loss processes," but that's a bit vague.

   RESPONSE: In an ideal world we would have been able to assess nitrogen loss processes following the upwelling of two much more distinct deep waters, in terms of their N-deficit. We had identified waters at two locations but, unfortunately, by the time we did collect them, their signatures were quite similar. Hence, the focus of this paper is to more generally 'quantify the importance of nitrogen loss processes in overall nitrogen cycling following simulated deep-water upwelling in the Humboldt Current System'. This is a first-time study and the unique closed-system mesocosm budgeting approach has revealed interesting conclusions that fit broader scale in-situ observations.

**2.** The paper has a lot about the nitrogen budget and about comparing the mesocosms to the real Pacific, but I think all that should be minimized. The mesocosms were contaminated by birds and the added 15N apparently stimulated rates.

   RESPONSE: It appears that there is a misunderstanding in regard to the mesocosm nitrogen budget and comparisons with the surrounding Pacific. The onset of the bird faeces contamination was day 40. Hence, we have restricted the budget calculations to the first 38 days of measurements and excluded the rest. Concerning 15N label stimulating measured nitrogen loss rates, in particular heterotrophic denitrification, this study is not the first to make such observation. This can happen at relatively low dissolved inorganic nitrogen and high organic matter availability. We did account for this observation (when making the direct comparison of the full nitrogen budget with rate measurements in the mesocosms and extrapolations to the Pacific) by using 'maximum sustainable rates', derived from in-situ dissolved inorganic nitrate concentrations rather than maximum attainable rates measured during incubations.

**3.** I think the authors should concentrate on comparing denitrification vs. anammox. As mentioned below in more detail, they don't address why their rates of anammox were low compared with previous studies and why anammox apparently was lower in the mesocosms than in the real ocean.

   RESPONSE: We will strengthen the two and a half pages of discussion of the denitrification vs. anammox findings (see detailed comments/responses below), and will also directly address this issue by adding the following statement in the abstract: 'Both in the mesocosms and the Pacific Ocean anammox made only a minor contribution to overall nitrogen loss when encountered, potentially related to organic matter C/N stoichiometry and/or process specific oxygen and hydrogen sulphide sensitivities.'

**Specific Comments**

**L8:** I think "actual" is better than "realized."

RESPONSE: We agree and will make the suggested change.

**L9:** I suggest removing the comparison of rates in the mesocosm with rates in the real ocean.

RESPONSE: Please see our response to general comment #2.

**L28:** Note the misspelling, "denitriciation."

RESPONSE: Thank you for pointing out this typo, it will be changed.

**L40:** Higher temperature explains oxygen loss in the upper water column, but only accounts for about half of the loss in deeper waters.

RESPONSE: We will add the following clarification: 'Furthermore, due to increasing temperatures the ocean looses oxygen ($O_2$) and OMZs are expanding (e.g. Bopp et al. (2002); Bograd et al. (2008); Stramma et al. (2008); Oschlies et al. (2017)). Together with changes to microbial activity, this modifies biogeochemical properties of upwelled waters including, next to $O_2$, carbonate chemistry speciation...'

**L43:** Missing a word like "waters."

RESPONSE: We will add the term 'deep waters'.

**L74:** and elsewhere: "$umolL^{-1}$" should be "$umol\ L^{-1}$"—a space between umol and L.

RESPONSE: Thank you for pointing out this oversight, we shall make the necessary changes.

**L83:** Rather than emphasizing N:P ratios, I think the authors should emphasize that the extreme condition had unmeasurable NO3 and NO2 and a more negative N* than the moderate condition.

RESPONSE: We will clarify the text by the following statement: 'However, both waters had a quite strong N-deficit ($N^*$), in comparison to a typical N/P of 16/1 required for phytoplankton growth (Redfield et al., 1963; Brzezinski, 1985), and will be referred to as 'low N/P' and 'very low N/P' treatments in the following (compare Tab. 1).'

**L93:** Rather than "aka DNRA", the authors should just define DNRA. It's defined much later in the paper, but it should be here when the abbreviation is first used.

RESPONSE: We will make the suggested change.

**L95:** Note the misspelling here, "failry." I will stop noting other misspelling that the spellchecker on Word or other word processing programs would catch. The authors should assume the journal won't do much copy editing.

RESPONSE: We apologise for yet another typo and will thoroughly check a revised version.

**L143:** Rather than "orni-eutrophication," I suggest "avian eutrophication."

RESPONSE: This term was introduced in the accompanying paper by Bach et al. (2020), hence we are inclined to keep it, for consistency.

**L180:** Fig A3 seems to be referred to before Fig A1 and A2, which is not standard practice.

RESPONSE: The order should be alright, as A1 and A2 are referred to before A3 on original L180.

**L204:** The authors emphasize that the "theoretical" sustainable rate of denitrification is based on changes in NO3+NO2 concentrations. But what about nitrification supplying NO3+NO2? The authors seem to imply nitrification didn't occur because of the lack of oxygen, but the gas was measurable, perhaps at levels high enough for nitrification.

RESPONSE: The reviewer makes an important point. Nitrification has indeed been found to operate at the low micro-molar (and even nano-molar) levels observed in our study (Bristow et al. 2016). And it appears that at such oxygen levels there is cyclic nitrogen turn-over by nitrite oxidation followed by nitrate reduction, not contributing to nitrogen loss via $N_2$ (Babbin et al. 2020), further complicating the picture. However, nitrification (ammonium oxidation) rates measured in Bristow et al. 2016, and by Peng et al. 2016 and Santoro et al. 2021 were usually at least an order of magnitude lower than measured denitrification rates in our study. Hence, it is unlikely that nitrification played a significant role in supplying nitrite for denitrification. We will add this piece of information to the discussion.

**L226:** The authors have a table and a very complex, four-panel figure (see below) about the multi-variable linear regression work, but all that is accompanied by two short paragraphs. That's an indication that the figure and the table are overkill. Readers will care only (if they do at all) about the best model, not the rest of the stuff given in the figure.

RESPONSE: We will simplify the figure by removing the second-best fit. We will also add to the discussion that the finding that the main drivers of denitrification were nitrite and organic matter availability suggest that heterotrophic denitrification rather than chemolithoautotrophic was the dominant N-loss process.

**L227 and elsewhere:** The authors shouldn't use "measured/maximum" because it's ambiguous. Which is it? The measured rate or the highest one? At the very least they should define what they mean, but I don't think the term should be used at all.

RESPONSE: We agree, this has been ambiguous. We will change to 'measured/maximum-sustainable rates', making clear that in cases where substrate limitation was encountered, maximum-sustainable rather than measured rates were used for in-situ N-loss estimates.

**L256:** I think it doesn't make sense that NO2- is more important than NO3- in driving denitrification. This is worth a brief explanation, perhaps.

RESPONSE: As denitrification from $NO_3^-$ to $N_2$ involves multiple and independent steps and organisms the correlation between $N_2$ production and a substrate concentration should become better the closer one gets to the end of this chain (Fig. 1). For example, there should be a perfect correlation between $N_2O$ concentrations and $N_2$ production, and $NO_3^-$ concentrations and their turn-over to $NO_2^-$ are meaningless if the intermediate steps to nitric and nitrous oxide are blocked or constitute a bottle-neck. This also explains the finding that nitrate reduction to nitrite often exceeds the total rate of denitrification to $N_2$.

**L303:** The authors end this section with textbook stuff about denitrification vs anammox with a generalization about which can be observed in the absence of the other. I think much of this can be deleted and replaced a more critical discussion of their data.

The authors need to grapple with the more important and novel findings from their study: that anammox wasn't as high as measured in previous studies and that it wasn't as high (I don't believe) in their mesocosms than in the real ocean.

RESPONSE: We agree, that this section ends with textbook knowledge. It is basically setting the stage for the in-depth discussion on what could explain our denitrification/anammox observations in the following sections. Hence, we are inclined to keep it.

The discussion that follows over the next two sections is actually an attempt to explain why anammox wasn't as high as in many previous studies. Finally, we agree that anammox rates were equally low in the mesocosms and the surrounding Pacific, which is highlighted in the abstract as 'Both in the mesocosms and the Pacific Ocean anammox made only a minor contribution to overall nitrogen loss when encountered...'

**L204:** Not picked up by a spell-checker: it should be "absence," not "absences."

RESPONSE: Thank you for picking up this typo.

**L297:** What do the authors mean by "anammox dominance"? They didn't see that, and the theoretical maximum contribution by anammox is only 28

RESPONSE: We will clarify our point here by changing the sentence to: 'The reason for an anammox dominance in several studies mentioned above,...'

**L306:** This section about organic matter C/N should be deleted. The authors found a typical Redfield ratio, but then spend several sentences arguing against their data. The entire paragraph doesn't add enough to the paper to be worth taking up space in the Discussion.

RESPONSE: We are not arguing against our data, but rather try to explain low anammox contributions to overall N-loss. High C/N ratios of organic matter being decomposed by denitrifiers would offer an explanation. And carefully examining the data at hand, a number of possibilities are identified why this might indeed have been the case.

**L381:** The paper ends very abruptly. I'm not a fan of ending papers with a summary, but it would be nice to see something about the implications of the authors' work for the Big Picture.

RESPONSE: We will add a more general final paragraph, reading: 'Nitrogen cycling in ODZs and OMZs currently plays a very important role in the overall marine nitrogen budget. However, the magnitude and direction of change in the actual nitrogen loss term in response to ongoing climate and ocean change (e.g. ocean stratification, acidification and/or changes in temperature and oxygen levels) is uncertain. This issue is further complicated by uncertainties

in future primary productivity and organic matter export. For instance, depending on the representative concentration pathway, future export production could decrease as a result of changes to community structure (see Bindhoff et al. (2019) for details and refs. therein). In summary, future changes in upwelling intensity and frequency, as well as the other potential biotic and abiotic factors mentioned above, could change the nitrogen (im)balance in ODZs and OMZs, having a significant impact on the overall marine nitrogen budget.'

**Table 1:** Note that NO2- has just one negative charge—it's not NO2⁼2.

RESPONSE: Thanks for spotting this typo.

**Table 2:** Data in this table can be used to make several comparisons, which complicates it: the moderate vs. extreme treatments, 15N rates vs. concentration changes, mesocosms vs the real ocean, and denitrification vs. anammox. I suggest the authors need to re-think the design of this table and use another format, break it up, or put some data in a figure.

If the table is kept, the formatting needs to be improved. Colors and ( ) to denote different types of data should be avoided because the main body of the table can't be understood without looking at the table caption, making the reader work harder than necessary.

I think it's important to give integrated rates for anammox vs. denitrification, so readers can evaluate how the two processes compared for the mesocosms versus the real ocean.

Finally, the overall average and its SD for all mesocosms and the Pacific Ocean are rather meaningless. The authors should report the average and error for the two types of mesocosms alone...

RESPONSE: We will re-format the table, as suggested by the reviewer, and remove colours, re-organise the mesocosms into treatments and calculated means and standard deviations separately to facilitate comparisons.

Concerning integrated individual rates of denitrification and anammox we have opted to sum both processes up as anammox was not encountered in most mesocosms and had only a minute contribution to overall N loss in the others.

**Table 3:** The caption should say that the regression analysis was done to explain the rate of denitrification.

RESPONSE: We will make the suggested change.

**Figure 1:** This figure is more appropriate for a textbook or a review paper, not this paper. It should be deleted. Maybe one of the figures now in supplemental materials, such as Fig A2, could be upgraded to the main paper.

RESPONSE: We are inclined to leave the figure in, as it is helpful in understanding certain aspects of the discussion, for example the rationale behind our response to the L256 comment. This will be particularly useful for a non-expert reader.

**Figure 3:** The authors should say explicitly that M1-M8 are mesocosms. "Bottom" in all of

the y-axis labels can be deleted and moved to the figure caption. The labels would be cleaner and easier to read.

RESPONSE: We will explicitly mention that M1-M8 refer to the various mesocosms. We are inclined, however, to keep 'Bottom' in the y-axis labels as a reader will immediately realise where samples were taken, without having to consult the figure caption.

**Figure 4:** Note that NH4- should be NH4+.

RESPONSE: Thanks for spotting this typo.

**Figure 5:** Most of this figure doesn't make sense to me, and it seems overkill. It should be deleted. A table summarizing the best model would suffice.

RESPONSE: We will streamline and simplify the figure.

**Figure A4** Explain the symbols and colors, etc. Don't force readers to work and go back to Figure 3.

RESPONSE: We will add a proper description of the colour-coding and symbols to the caption of figure A3 and then refer to it, i.e. all necessary information will be contained in the Appendix.

---

## Referee Report (RR1)

**General comments**
The authors addressed some of my concerns during their revisions. However, I am still a bit dissatisfied with the quality of their oxygen measurements, which should have been more carefully done *in-situ*, and also during their incubations. Denitrification and anammox are highly dependent on oxygen concentrations, even at nanomolar levels. I do not quite understand why they would observe such a high offset (+13 $\mu mol\ L^{-1}$) between Winkler titrations and their optical CTD sensor. Were Winkler titrations performed at all depths? Oxygen concentrations are such important measurements for their study, it is a pity that more attention wasn't dedicated to calibrating their CTD sensor better. Second, I would like to see a more thorough assessment of how important gases ($O_2$, $H_2S$) influence nitrogen loss. They added a short discussion in their manuscript, but perhaps back of the envelope calculations would be more helpful to understand the effect of purging with He on the concentration of these gases. In the case of $H_2S$, a reduction of 80% after purging is quite a lot and is most likely to affect chemoautotrophic nitrogen loss ($H_2S$ oxidation coupled to $NO_3^-$ reduction). As was pointed out by reviewer #2, I think the authors need to be more careful in interpreting these rates at the larger scale, especially since it was recognized that rates were stimulated due to substrate limitation during the $^{15}N$-labeled incubations. These are **potential** rates that provide information about processes; hence these rates should be interpreted with caution and the limitations of this study should be better acknowledged. I agree with reviewer 2 that the authors should focus on the importance of different processes (denitrification versus anammox) rather than trying to compare their rates to the full nitrogen budget in the mesocosms (any strong agreement between the two is likely fortuitous). I also agree with reviewer #2 that $NO_2^-$ cannot be deemed to be "more important" than $NO_3^-$ for denitrification. The substrate addition during $^{15}N$-labeled incubations relative to *in-situ* concentrations would generally be higher for $NO_2^-$ than $NO_3^-$, which could further stimulate rates. The authors need to cite relevant study that observed clear discrepancies between $^{15}N$-labeled rate measurements using both $NO_2^-$ and $NO_3^-$ and perhaps briefly discuss these differences. On this note, I do not think the authors understood my argument regarding the production of excess $^{29}N_2$ during $^{15}N$-$NO_2^-$ labeled incubations (see Chang et al. 2019 paper). This process needs to be discussed in the manuscript (in connection with their data). Finally, there were quite a few typos in the original version of the manuscript and quite a few typos also were introduced during revisions (see below). The authors should be more careful when revising their manuscript.
Despite all these concerns, I still think that this is an interesting and novel study, as few such mesocosm experiment exist (if at all) for the ETSP ODZ.

**Specific comments:**

Line 40: Adding "of deep waters" after frequency does not make sense as they are referring to upwelling frequency.

Lines 46-48: at least one reference needs to be added at the end of this sentence.

Lines 63-64: What were measured $H_2S$ concentrations using this sensor for incubated waters? That could help address the He purging issue if $H_2S$ concentrations were negligible.

Lines 77-82: This section is still a bit vague. Oxygen concentration is very important in controlling nitrogen loss rates. Were oxygen concentrations also measured at all depths using Winkler titrations? If so, I think these values should be used instead if they experienced issues with their CTD sensor calibration. I don't think it is sufficient to say: "Hence oxygen concentrations …. are likely to have been significantly lower". How much lower? Are they sure that conditions were truly anoxic, and conducive to nitrogen loss?

Line 86: Remove "However" at the beginning of sentence.

Line 103-106: Also cite the new manuscript by Bourbonnais et al. (2020) in Frontiers that describes these types of incubations in detail as well as provide calculation templates: Bourbonnais, A, C. Frey, X. Sun, L. A. Bristow, A. Jayakumar, N. E. Ostrom, K. L. Casciotti, and B. B. Ward. (2021), Protocols for assessing transformation rates of nitrous oxide in the water column, *Frontiers in Marine Science* 8, 293.

Lines 102-103: I think this sentence is ambiguous. Change to "Our calculations have shown that exchanging the bottle volume at least 24 times is required to reduce the $O_2$ concentration to less than 20% of *in-situ* conditions.

Line 103: "Observered" is misspelled!

Lines 113-115: This sentence is a bit vague. Could they provide a back of the envelope calculation to better estimate how He purging would affect $H_2S$ concentrations? This is important to assess the role of chemoautotrophic versus heterotrophic denitrification.

Lines 118-122: This sentence is too long – I suggest breaking in two.

Line 125-131: Why was $^{30}N_2$ noisy? I think that their rates are high enough to get a good $^{30}N_2$ signal. Calculating denitrification rates using mass 29 can be problematic as other studies reported production of excess $^{29}N_2$ that could not be accounted by assuming binomial distribution (after considering the contribution from anammox). I strongly recommend the authors to read de Brabandere et al. (2013) and Chang et al. (2014) for more information regarding this process. These authors attributed the production of excess $^{29}N_2$ during $^{15}N$-$NO_2^-$ incubations to "nitrite shunting" where unlabeled $NO_3^-$ is converted to $N_2$ completely intracellularly without exchange with the external ambient $NO_2^-$ pool. The paper by Chang et al. (2019) needs to be discussed/cited.
Chang, B. X., Rich, J. R., Jayakumar, A., Naik, H., Pratihary, A. K., Keil, R. G., ... & Devol, A. H. (2014). The effect of organic carbon on fixed nitrogen loss in the eastern tropical South Pacific and Arabian Sea oxygen deficient zones. *Limnology and oceanography*, *59*(4), 1267-1274.

Lines 168 and 177: Remove the word "please"

Lines 226-230: I think this similarity is fortuitous since only **potential** rates are measured using $^{15}N$-labeled incubations.

Lines 261-263: I do not think complete nitrogen loss occurs at such high $O_2$ concentrations (30-40 $\mu$mol $L^{-1}$).

Lines 269-278: It would have been best to construct the Michaelis-Menten curves as in Michiels et al. (2019) or use published Michaelis-Menten parameters (for the same or similar environments – as published in Michiels et al. (2019)) to estimate denitrification rates at *in-situ* $NO_2^-/NO_3^-$ concentrations (rather than using a maximum nitrogen loss rates based on nutrient concentrations). The authors did not well address this point in their response to my previous review.

Line 286-287: Would that rather be an upper boundary estimate (relative to true environmental conditions), since orni-eutrophication and using maximum-sustainable denitrification rates would artificially increase their nitrogen loss rate estimates?

Line 343: Change to "over-consumption"

Lines 364-375: The authors need to acknowledge that purging with He before their $^{15}N$-labeled incubations would reduce the $H_2S$ concentrations. Hence, these rates should be interpreted with caution.

---

## Author Response (AR2)

**Response to Reviewer #1**

We thank Reviewer #1 for their insightful comments and suggestions. Below please find our detailed point-by-point response.

**General Comments**

**1.** 'The authors revised their paper in response to many of the reviewers' comments, but they didn't really try to address with my main complaint. Why this study? They still have only a rather vague statement about wishing "to quantify the importance of nitrogen loss processes in overall nitrogen cycling." Maybe a more precise question is, why do the mesocosms? ...'

RESPONSE: As suggested, we have expanded on the aims of the mesocosm experiment (please see our response to the Line 48 comment below). We furthermore added '– a mesocosm approach' to the title, to highlight the complimentary novelty of our experiment.

**Specific comments of Reviewer #1**

**Line 9:** The argument why substrate limitation explains the difference is not clear. See below my comment on L265.

RESPONSE: There is probably a misunderstanding here i.e. measured and potentially realised rates in the mesocosms and Pacific were similar. Substrate limitation is most likely to explain that measured rates were higher than theoretically sustainable in the mesocosms. The confusion might have been caused by using the term in-situ, to distinguish processes in the mesocosms from those during bottle incubations. We have now changed the respective sentence to 'However, actual in-situ rates in the mesocosms, estimated via Michaelis-Menten kinetic scaling, did most likely not exceed 0.2–4.2 $\mathrm{nmol\ N_2\ L^{-1}h^{-1}}$ (interquartile range), due to substrate limitation.'

**Line 33:** I'm not sure what this means: "the net balance in terms of bioavailable nitrogen is negative." All of the processes discussed before this sentence are about how N is lost, so of course, they are "negative." Why is this sentence needed?

RESPONSE: We agree, and have removed the sentence.

**Line 48:** The authors here could add a couple of sentences about what they hope to learn from their mesocosm experiment.

RESPONSE: As suggested, we have changed the last paragraph of the introduction to: ' To better understand the events following the coastal upwelling of oxygen and nitrogen depleted deep waters, we make use of an off-shore mesocosm setup. Such approach allows simulating upwelling and tracing biogeochemical element cycling and associated trophic interactions. We were specifically aiming to address the question of nitrogen cycling, i.e. the build-up and turnover of organic nitrogen pools, their export from the surface to depth, and most importantly, potential loss processes. Because such approach enables budgeting of the various pools, it will be an alternative and independent assessment of the nitrogen balance in coastal ODZs, next to classical shipboard transects.'

**Line 84:** The N/P ratio, to be picky, isn't given in Table 1, just N/P/Si. More obvious than the difference in N/P ratios between the two waters is the presence or absence of NO3 and NO2. That's seen in the first two columns of Table 1. I think it's a better way to label the two waters.

RESPONSE: The N/P/Si ratio given in Table 1 allows to directly read off the N/P ratio. The reason why we would prefer to keep the 'low N/P' and 'very low N/P' terminology (besides that it appears valid, with 1.7/1 and 0.1/1), is that it will be consistent with the accompanying paper by Bach et al. 2020.

**Line 265:** If denitrification were substrate-limited in the mesocosms, wouldn't you expect rates to be lower, not higher than in situ? The authors' argument is not clear.

RESPONSE: This is probably the same misunderstanding as above, i.e. in-situ is used not to describe conditions outside the mesocosms but within, contrasting the lab incubations. We hope to clarify this by changing to :'Most interestingly, measured denitrification rates in incubations of mesocosm waters during the second half of the experiment exceeded those theoretically sustainable at in-situ substrate availability.'

**Line 270:** The authors changed "measured/maximum" to "measured/maximum sustainable rates", in response to my comment that it's ambiguous. In their response to the reviewers, they say they made the change to make "clear that in cases where substrate limitation was encountered, maximum-sustainable rather than measured rates were used for in-situ N-loss estimates." But that's not clear in the Discussion. "measured/maximum sustainable rates" is still ambiguous by itself. Use of "/" often leads to ambiguity.

RESPONSE: We have adopted the suggestion by reviewer #2 to use Michaelis-Menten kinetic scaling, rather than maximum sustainable rates derived from substrate concentrations, to estimate in-situ from measured rates in incubations. This should avoid any confusion.

**Line 279:** The authors say inorganic N didn't differ much between the two mesocosms, citing Table 1. But that table has fairly high NO3 and NO2 ($>$1 uM) for one mesocosm and zero NO3 and NO2 for the other. That seems like a big difference to me. Maybe over time, the difference disappeared?

RESPONSE: We state that the differences in dissolved inorganic nitrogen between these water masses are relatively small in relation to the overall nitrogen pool in the mesocosms. In other words, adding 3 $\mu$molL$^{-1}$ of DIN to a 30 $\mu$molL$^{-1}$ background of bioavailable nitrogen will change the overall nitrogen, potentially available for denitrification and anammox, by only 10%, as opposed to no addition.

**Table 2**: The author revised this table in response to most of my criticisms. I still think it would be better if they label the rows in the table, not give explanations in the caption, to indicate the expected maximum rates and the anammox values. I had to think too much to interpret the * and the italics used to indicate the anammox values. What's the "overall mean"? It needs to be explained.

RESPONSE: We actually had added '*anammox**'* to the row label, as suggested. We have

changed 'Mean' to 'Treatment Mean', which should clarify that the 'Overall Mean' is across all mesocosms. Finally, the table was further simplified as now only reporting in-situ rates, and measured rates being presented in a separate table in the appendix.

Also, I suggest putting a "-" in front of the N-loss estimates (make them negative—they are losses) to make clear they can be compared directly with the N-budget numbers.

RESPONSE: Done.

Table 3 and Figure 5: Although the authors have simplified Table 3, I still think it and Figure 5 aren't needed. The authors don't really use these to make any points. I still don't understand Figure 5.

RESPONSE: We actually do use information provided in this table and figure to make an important argument, i.e. 'The primary drivers of in-situ denitrification rates appeared to be the availability of $NO_2^-$ and $NO_3^-$, followed by particulate organic matter (nitrogen), as indicated by multiple linear regression and effect size analysis (Fig. 5). This suggests that heterotrophic rather than chemolithotrophic denitrification was dominant (compare section 4.2.2), as both are substrates for the former process and eventually limit rates.'

Concerning Figure 5, what might be confusing is our approach to not simply use all seven measured potential predictors for an MLR. In order to find the sweet spot between model complexity (number of predictors in the model out of the seven possible) and avoid overfitting, we employed multiple stepwise MLRs. To explain the rational and benefit we have added the following to the figure caption: 'The red arrows denote the relatively small increase in maximum $R^2$ beyond 5 main variables and the relatively large decrease below, indicating this to be the sweet-spot in terms of balancing model complexity with predictive power, avoiding overfitting (see Methods for details).' We have also slightly modified a sentence in the methods section, hopefully providing more clarity on the approach.

**Response to Reviewer #2**

We thank Reviewer #2 for their insightful comments and suggestions. Below please find our detailed point-by-point response.

**General Comments**

**1.** ... However, I am still a bit dissatisfied with the quality of their oxygen measurements, which should have been more carefully done in-situ, and also during their incubations. Denitrification and anammox are highly dependent on oxygen concentrations, even at nanomolar levels. I do not quite understand why they would observe such a high offset (+13 $\mu$molL$^{-1}$) between Winkler titrations and their optical CTD sensor. Were Winkler titrations performed at all depths? Oxygen concentrations are such important measurements for their study, it is a pity that more attention wasn't dedicated to calibrating their CTD sensor better.

RESPONSE: After factory calibration prior to the experiment, the optical oxygen sensor was two point field calibrated as per the manufacturer's instructions (for 0 and 100%) every other deployment. The most likely explanation for the offset to the two Winkler samples that

were measured is the extremely steep oxycline, basically going from 100% at the surface to almost 0% within 10 to 15 metres. Hence, any sensor delay will cause the probe to read higher oxygen concentrations than actually present. That is the reason why we employed a hysteresis correction, assuming a response time of 1 second (based on information provided by the manufacturer). However, if the response time was actually 2 to 2.5 seconds, an offset of $13 \mu molL^{-1}$ would be explained. So, the most likely explanation is that the response time of the optical oxygen sensor was actually slower than we thought. We have added this piece of information to the methods section. In hindsight, we should have measured the actual hysteresis of our sensor, as well as taken more Winkler samples. The realisation that getting precise oxygen concentration measurements from CTD casts when gradients are extremely steep, only materialised when carefully analysing the data after the experiment.

**2.** ... Second, I would like to see a more thorough assessment of how important gases ($O_2$, $H_2S$) influence nitrogen loss. They added a short discussion in their manuscript, but perhaps back of the envelope calculations would be more helpful to understand the effect of purging with He on the concentration of these gases. In the case of $H_2S$, a reduction of 80% after purging is quite a lot and is most likely to affect chemoautotrophic nitrogen loss ($H_2S$ oxidation coupled to $NO_3^-$ reduction).

RESPONSE: There is a misunderstanding here, and we have changed our wording in the Methods section to be more clear about it. The He flow comparison with Holtappels et al. suggests that the reduction of $O_2$ was not to less than 20% of in-situ concentrations but by less than 20%. Hence, that is also the case for all the other gases, and even more so for $H_2S$, as it is buffered. Hence, given the relatively high $H_2S$ concentrations, potential chemolithotrophic denitrification should not have been limited by $H_2S$. In fact, measured denitrification was best explained by nitrite, nitrate and particulate organic nitrogen concentrations, and $H_2S$ had hardly any explanatory power (see MLR results), suggesting that chemolithotrophic denitrification was a minor contributor, if any.

**3.** As was pointed out by reviewer #2, I think the authors need to be more careful in interpreting these rates at the larger scale, especially since it was recognized that rates were stimulated due to substrate limitation during the 15 N-labeled incubations. These are potential rates that provide information about processes; hence these rates should be interpreted with caution and the limitations of this study should be better acknowledged. I agree with reviewer 2 that the authors should focus on the importance of different processes (denitrification versus anammox) rather than trying to compare their rates to the full nitrogen budget in the mesocosms (any strong agreement between the two is likely fortuitous).

RESPONSE: As already pointed out in our previous response to reviewer #1 comments, the overall nitrogen loss over the first 38 days of the experiment (prior to orrieutrophication) had been calculated using maximum sustainable rates, when substrate limitation was encountered. Hence, the finding that the addition of labelled $NO_2^-$ stimulated measured rates, especially during the second half of the experiment, had been removed from for the overall nitrogen loss estimate. We have now also adopted the suggested approach to estimate in-situ rates from Michaelis-Menten kinetic scaling (see below). Together with the fact that there is an overall

good agreement of measured N loss with the full nitrogen budget, lends confidence to interpret findings in a larger context. And as we have pointed out, there is no statistically significant correlation between individual measured N loss and the full nitrogen budget, which would indeed be fortuitous given the number of entities in the mass balance. However, overall mean N loss and N budget are indistinguishably close, again, lending confidence to our findings and resulting conclusions.

**4.** I also agree with reviewer #2 that $NO_2^-$ cannot be deemed to be "more important" than $NO_3^-$ for denitrification.... The authors need to cite relevant study that observed clear discrepancies between $^{15}N$-labeled rate measurements using both $NO_2^-$ and $NO_3^-$ and perhaps briefly discuss these differences.

RESPONSE: We have added the follow explanation and reference to the manuscript: 'The reason for $NO_2^-$ rather than $NO_3^-$ concentrations explaining rates of denitrification in one of the MLRs could be found in the following: As denitrification from $NO_3^-$ to $N_2$ involves multiple and independent steps and organisms, the correlation between $N_2$ production rate and a substrate concentration should become better the closer one gets to the end of this chain (Fig. 1). For example, there should be a perfect correlation between $N_2O$ concentrations and $N_2$ production rate, and $NO_3^-$ concentrations and their turn-over to $NO_2^-$ are meaningless if the intermediate steps to nitric and nitrous oxide are blocked or constitute a bottle-neck. This also would contribute to the finding that denitrification rate measurements based on $^{15}NO_3^-$ can be lower than those based on $^{15}NO_2^-$ (Hamersley et al. 2007).

**5.** On this note, I do not think the authors understood my argument regarding the production of excess $^{29}N_2$ during $^{15}N$-$NO_2^-$ labeled incubations (see Chang et al. 2019 paper). This process needs to be discussed in the manuscript (in connection with their data).

RESPONSE: We have added the following statement to the methods section: 'It is noted that there have been studies which found discrepancies between denitrification calculated using $^{29}N_2$ as above, and $^{30}N_2$, due to non-binomial distributions (De Brabandere et al., 2014; Chang et al., 2014). This has been attributed to so-called intra-cellular '$NO_3^-$/$NO_2^-$-shunting', which leads to an error in the calculations of labelled to unlabelled substrate, as based on known additions and measured seawater concentrations. As of noisy $^{30}N_2$ data, we can not check if that was an issue here, yet it would lead to an underestimation of denitrification rates in both cases. Given the good agreement between our rate measurements and a full nitrogen budget (see section 3.2 for details), however, it appears that potential '$NO_3^-$/$NO_2^-$-shunting' did not affect our rate measurements significantly.'

**Specific Comments**

**Line 40:** Adding "of deep waters" after frequency does not make sense as they are referring to upwelling frequency.

RESPONSE: It is both, the upwelling intensity and upwelling frequency of deep waters.

**Line 46-48:** At least one reference needs to be added at the end of this sentence.

RESPONSE: We have changed 'can' into 'could' as there are no studies to date yet.

**Line 63-64:** What were measured $H_2S$ concentrations using this sensor for incubated waters? That could help address the He purging issue if $H_2S$ concentrations were negligible.

RESPONSE: The CTD sensor is too large to fit into the 12 ml Exetainers, hence $H_2S$ could not be measured directly for the incubations. Please see also our response the the reviewer's general comment #2.

**Line 77-82:** This section is still a bit vague. Oxygen concentration is very important in controlling nitrogen loss rates. Were oxygen concentrations also measured at all depths using Winkler titrations? If so, I think these values should be used instead if they experienced issues with their CTD sensor calibration. I don't think it is sufficient to say: "Hence oxygen concentrations .... are likely to have been significantly lower". How much lower? Are they sure that conditions were truly anoxic, and conducive to nitrogen loss?

RESPONSE: We have added a likely explanation for the observed off-set (please also see our response to the reviewer's general comment #1). This also should clarify by how much, i.e. exactly the off-set.

**Line 86:** Remove "However" at the beginning of sentence.

RESPONSE: The 'However' makes sense, as having very similar deep water masses in terms of N-deficit goes against our intent described a sentence earlier, i.e. to simulate up-welling of deep water with two distinct OMZ signatures.

**Line 103-106:** Also cite the new manuscript by Bourbonnais et al. (2020) in Frontiers that describes these types of incubations in detail as well as provide calculation templates: Bourbonnais, A, C. Frey, X. Sun, L. A. Bristow, A. Jayakumar, N. E. Ostrom, K. L. Casciotti, and B. B. Ward. (2021), Protocols for assessing transformation rates of nitrous oxide in the water column, Frontiers in Marine Science 8, 293.

RESPONSE: Done.

**Lines 102-103:** I think this sentence is ambiguous. Change to "Our calculations have shown that exchanging the bottle volume at least 24 times is required to reduce the $O_2$ concentration to less than 20% of in-situ conditions.

RESPONSE: It is a reduction BY less than 20%, not TO less than 20%. We have clarified the text accordingly.

**Line 103:** "Observered" is misspelled!

RESPONSE: Thanks for finding this typo.

**Lines 113-115:** This sentence is a bit vague. Could they provide a back of the envelope calculation to better estimate how He purging would affect $H_2S$ concentrations? This is important to assess the role of chemoautotrophic versus heterotrophic denitrification.

RESPONSE: Please see our response to the reviewer's general comment #2.

**Lines 118-122:** This sentence is too long – I suggest breaking in two.

RESPONSE: We have split it in two.

**Line 125-131:** Why was $^{30}N_2$ noisy? I think that their rates are high enough to get a good $^{30}N_2$ signal. Calculating denitrification rates using mass 29 can be problematic as other studies reported production of excess 29 N 2 that could not be accounted by assuming binomial distribution (after considering the contribution from anammox). I strongly recommend the authors to read de Brabandere et al. (2013) and Chang et al. (2014) for more information regarding this process....

RESPONSE: Why it was noisy, we do not know. For the other concerns raised, please see our responses above.

**Lines 168 and 177:** Remove the word "please"

RESPONSE: Done.

**Lines 226-230:** I think this similarity is fortuitous since only potential rates are measured using $^{15}N$-labeled incubations.

RESPONSE: Please see our response to the reviewer's general comment #3 above.

**Lines 261-263:** I do not think complete nitrogen loss occurs at such high $O_2$ concentrations (30-40 $\mu molL^{-1}$).

RESPONSE: We simply cite the findings by Farías et al. 2009 here. We also do not speculate on whether the nitrogen loss would be 'complete'.

**Lines 269-278:** It would have been best to construct the Michaelis-Menten curves as in Michiels et al. (2019) or use published Michaelis-Menten parameters (for the same or similar environments – as published in Michiels et al. (2019)) to estimate denitrification rates at in-situ $NO_2^-$/nitrate concentrations (rather than using a maximum nitrogen loss rates based on nutrient concentrations). The authors did not well address this point in their response to my previous review.

RESPONSE: We thank the reviewer for bringing this up again, and totally agree. We have now calculated in-situ denitrification as suggested, using the half-saturation rate constant by Michiels et al. (2009), which is the highest published for water column denitrification. Hence, this conservative approach should rather under- than overestimate in-situ rates. We have also updated Tab. 2 accordingly, which furthermore simplified it, and added a new table to the appendix with the measured rates. We also re-ran the MLRs, using in-situ denitrification rates. Overall, the new in-situ rates are slightly lower than previously estimated, but all our conclusions remain.

**Line 286-287:** Would that rather be an upper boundary estimate (relative to true environmental conditions), since orni-eutrophication and using maximum-sustainable denitrification

rates would artificially increase their nitrogen loss rate estimates?

RESPONSE: The N-loss from the budget approach would underestimate losses if orni-eutrophication would have increased nitrogen pools in the mesocosms prior to T38. However, as there is no indication that this was the case, as well as it appeared to be confusing, we removed this statement.

**Line 343:** Change to "over-consumption"

RESPONSE: Done.

**Lines 364-375:** The authors need to acknowledge that purging with He before their 15 N-labeled incubations would reduce the $H_2S$ concentrations. Hence, these rates should be interpreted with caution.

RESPONSE: Please see our detailed response to the reviewer's general comment #2.